# A Topology-aware Graph Coarsening Framework for Continual Graph Learning

## Abstract

Continual learning on graphs tackles the problem of training a graph neural network (GNN) where graph data arrive in a streaming fashion and the model tends to forget knowledge from previous tasks when updating with new data. Traditional continual learning strategies such as Experience Replay can be adapted to streaming graphs, however, these methods often face challenges such as inefficiency in preserving graph topology and incapability of capturing the correlation between old and new tasks. To address these challenges, we propose (**TA**ℂℂ𝕆), a topology-aware graph coarsening and continual learning framework that stores information from previous tasks as a reduced graph. At each time period, this reduced graph expands by combining with a new graph and aligning shared nodes, and then it undergoes a "zoom out" process by reduction to maintain a stable size. We design a graph coarsening algorithm based on node representation proximities to efficiently reduce a graph and preserve topological information. We empirically demonstrate the learning process on the reduced graph can approximate that of the original graph. Our experiments validate the effectiveness of the proposed framework on three real-world datasets using different backbone GNN models.

## 1 Introduction

Deep neural networks are known to be oblivious: they lack the ability to consider any pre-existing knowledge or context outside of the information they were trained on. In offline settings, this problem can be mitigated by making multiple passes through the dataset with batch training. However, in a continual learning setup (also known as incremental learning or lifelong learning) (Tang & Matteson, 2020; Kirkpatrick et al., 2017; Rusu et al., 2016; Rolnick et al., 2019), this problem becomes more intractable as the model has no access to previous data, resulting in drastic degradation of model performance on old tasks. A variety of methods have been proposed to tackle the issue of "catastrophic forgetting". However, most of these approaches are tailored for Euclidean data, such as images and texts, which limits their effectiveness on non-Euclidean data like graphs. In this paper, we investigate continual graph learning (CGL) frameworks to address the forgetting problems in downstream tasks such as node classification on streaming graphs. Existing common CGL methods can be broadly categorized into *regularization*-based methods, *expansion*-based methods, and *rehearsal*-based methods. *Regularization*-based methods reserve old knowledge by adding regularization terms in training losses to penalize the change of model parameters (Liu et al., 2020) or the shift of node embeddings (Su & Wu, 2023). However, they are generally vulnerable to the "brittleness" problem (Tang & Matteson, 2020; Pan et al., 2020) that restrains the models' ability to learn new knowledge for new tasks. Expansion-based methods Zhang et al. (2023) that assign isolated model parameters to different tasks are inherently expensive. In this work, we consider *rehearsal*-based methods (Rolnick et al., 2019; Chaudhry et al., 2018b; 2019; Tang & Matteson, 2020; Lopez-Paz & Ranzato, 2017) which use a limited extra memory to replay samples of old data to a learning model. Although there are several attempts (Kim et al., 2022; Zhou et al., 2020; Wei et al., 2022; Zhang et al., 2022a) to use memory buffers for saving graph samples in the rehearsal process of online graph training. They often face challenges such as inefficiency in employing topological information of graph nodes and ineffectiveness in capturing inter-task correlations.

In this work, we focus on preserving the topological information of graph nodes with efficient memory in continual learning frameworks to improve downstream task performance. Downstream tasks play a pivotal role in many real-world applications. We notice that most existing research on CGL (Liu

et al., 2020; Zhou et al., 2020; Zhang et al., 2022a) focuses on either *task-incremental-learning (task-IL)* (van de Ven & Tolias, 2019) or a *tranductive* (Zhang et al., 2022b; Carta et al., 2021) setting , where the sequential tasks are independent graphs containing nodes from non-overlapping class sets. In this setting, the model only needs to distinguish the classes included in the current task. For instance, if there are 10 classes in total, and this experimental setting divides these classes into 5 tasks. Task 1 focuses on classifying classes 1 and 2, while task 2 classifies classes 3 and 4, and so on. In this paper, we aim to tackle a more realistic *inductive* and *Generalized Class-incremental-learning (generalized class-IL)* (Mi et al., 2020) setting . Real-world graphs often form in an evolving manner, where nodes and edges are associated with a time stamp indicating their appearing time, and graphs keep expanding with new nodes and edges. For instance, in a citation network, each node representing a paper cites (forms an edge with) other papers when it is published. Each year more papers are published, and the citation graph also grows rapidly. In such cases, it is necessary to train a model incrementally and dynamically because saving or retraining the model on the full graph can be prohibitively expensive in space and time. So we split a streaming graph into subgraphs based on time periods and train a model on each subgraph sequentially for the node classification task. In such cases, subgraphs are correlated with each other through the edges connecting them (e.g. a paper cites another paper from previous time stamps). The structural information represented by edges may change from previous tasks (*inductive*). Also, since the tasks are divided by time periods instead of class labels, new tasks could contain both old classes from previous tasks and new classes, so the model needs to perform classification across all classes (*generalized class-IL*). Unlike traditional class-incremental setting (van de Ven & Tolias, 2019), where tasks often have entirely distinct class sets, *generalized class-IL* allows same classes to be shared across tasks, and previously encountered classes may reappear in new tasks.

We further demonstrate the necessity of preserving old knowledge when learning on a streaming graph. We take the Kindle e-book co-purchasing network (He & McAuley, 2016; McAuley et al., 2015) as an example. We split the graph into 5 subgraphs based on the first appearing time of each node (i.e., an e-book). We observe a gradual shift of node class distribution over time, as shown in Figure 1(a). Furthermore, even for nodes in the same class, their features and neighborhood patterns can shift (Kim et al., 2022). Also, in real-life situations, tasks may have different class sets (e.g. new fields of study emerge

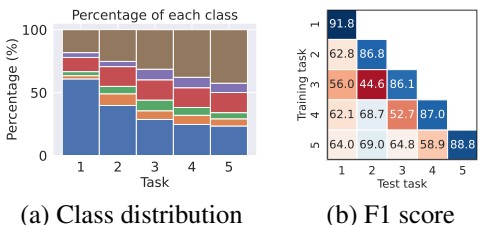

(a) Class distribution     (b) F1 score

Figure 1: A motivating example on Kindle dataset

and old fields of study become obsolete), which could exacerbate the forgetting problem. The F1 scores of node classification tasks using a Graph Convolutional Network (GCN) (Kipf & Welling, 2016) show that the model performs significantly worse on previous tasks when trained on new ones without any strategies to address the forgetting problem, as shown in Figure 1(b).

Motivated by these observations, we point out that despite the emergent research on Continual Graph Learning, two challenges still remain: 1) How to effectively and efficiently preserve the topological information of old tasks. 2) How to capture and take advantage of the correlation between old tasks and new tasks. In this work, we aim to address these two challenges by the following contributions.

- **A realistic CGL scheme.** We consider a realistic continual learning paradigm on streaming graphs. In this online setting, the model (i.e., a GNN) is trained on a subgraph (i.e., task) comprising edges from the current time period.
- **A dynamic CGL framework.** We propose a continual graph learning framework, **TA**$\mathbb{CO}$, that is efficient in memory and highly modular. By compressing a streaming graph online, **TA**$\mathbb{CO}$ addresses the challenges of preserving topological information of nodes and taking into account the correlation between old and new tasks due to overlapping nodes among subgraphs. **TA**$\mathbb{CO}$ does not introduce new learnable parameters besides the backbone GNN model.
- **A scalable graph reduction approach.** As a component of the CGL framework, we propose an efficient graph reduction algorithm, **RePro**, which utilizes both graph spectral properties and node features to efficiently reduce the sizes of input graphs. Additionally, we present a strategy, *Node Fidelity Preservation*, to ensure that certain nodes are not compressed, thereby maintaining the

quality of the reduced graph. We theoretically prove that Node Fidelity Preservation can mitigate the problem of vanishing minority classes in the process of graph reduction.

- **Evaluation.** We conduct extensive experiments and perform comprehensive ablation studies to evaluate the effectiveness of **TA**ℂ**O** and **RePro**. We also compare our method with multiple state-of-the-art methods for both CGL and graph coarsening tasks.

## 2 RELATED WORK

**Catastrophic forgetting.** *Regularization*, *Expansion*, and *Rehearsal* are common approaches to overcome the catastrophic forgetting problem (Tang & Matteson, 2020) in continual learning. *Regularization* methods (Kirkpatrick et al., 2017; Chaudhry et al., 2018a; Zenke et al., 2017; Ebrahimi et al., 2019) penalize parameter changes that are considered important for previous tasks. Although this approach is efficient in space and computational resources, it suffers from the "brittleness" problem (Tang & Matteson, 2020; Pan et al., 2020) where the previously regularized parameters may become obsolete and unable to adapt to new tasks. *Expansion*-based methods (Rusu et al., 2016; Yoon et al., 2017; Jerfel et al., 2018; Lee et al., 2020) assign isolated model parameters to different tasks and increase the model size when new tasks arrive. Such approaches are inherently expensive, especially when the number of tasks is large. *Rehearsal*-based methods consolidate old knowledge by replaying the model with past experiences. A common way is using a small episodic memory of previous data or generated samples from a generative model when it is trained on new tasks. While generative methods (Achille et al., 2018; Caccia et al., 2019; Deja et al., 2021; Ostapenko et al., 2019) may use less space, they also struggle with catastrophic forgetting problems and over-complicated designs (Parisi et al., 2018). Experience replay-based methods (Rolnick et al., 2019; Chaudhry et al., 2018b; 2019; Tang & Matteson, 2020; Lopez-Paz & Ranzato, 2017), on the other hand, have a more concise and straightforward workflow with remarkable performance demonstrated by various implementations with a small additional working memory.

**Continual graph learning.** Most existing CGL methods adapt *regularization*, *expansion*, or *rehearsal* methods on graphs. For instance, Liu et al. (2020) address catastrophic forgetting by penalizing the parameters that are crucial to both task-related objectives and topology-related ones. Su & Wu (2023) mitigate the impact of the structure shift by minimizing the input distribution shift of nodes. Rehearsal-based methods (Kim et al., 2022; Zhou et al., 2020; Wei et al., 2022) keep a memory buffer to store old samples, which treat replayed samples as independent data points and fail to preserve their structural information. Zhang et al. (2022a) preserve the topology information by storing a sparsified $L$-hop neighborhood of replay nodes. However, storing topology information of nodes through this method is not very efficient and the information of uncovered nodes is completely lost; also it fails to capture inter-task correlations in our setup. Besides, Feng et al. (2023) present an approach to address both the heterophily propagation issue and forgetting problem with a triad structure replay strategy: it regularizes the distance between the nodes in selected closed triads and open triads, which is hard to be categorized into any of the two approaches. Some CGL work (Wang et al., 2021; Xu et al., 2020) focuses on graph-level classification where each sample (a graph) is independent of other samples, whereas our paper mainly tackles the problem of node classification where the interdependence of samples plays a pivotal role. It is worth noting that **dynamic graph learning** (DGL) (Wang et al., 2020; Polyzos et al., 2021; Ma et al., 2020; Feng et al., 2020; Bielak et al., 2022; Galke et al., 2020) focuses on learning up-to-date representations on dynamic graphs instead of tackling the forgetting problem. They assume the model has access to previous information. The challenges explored in the DGL differ fundamentally from those addressed in our paper.

**Graph coarsening.** Scalability is a major concern in graph learning. Extensive studies aim to reduce the number of nodes in a graph, such that the coarsened graph approximates the original graph (Jin et al., 2020; Loukas, 2018; Chen & Safro, 2011; Livne & Brandt, 2012). In recent years, graph coarsening techniques are also applied for scalable graph representation learning (Liang et al., 2018; Deng et al., 2019; Fahrbach et al., 2020; Huang et al., 2021) and graph pooling (Pang et al., 2021; Liu et al., 2023; Lee et al., 2019) . Most graph coarsening methods (Jin et al., 2020; Loukas, 2018; Chen & Safro, 2011; Livne & Brandt, 2012) aim to preserve certain spectral properties of graphs by merging nodes with high spectral similarity. However, such approaches usually result in high computational complexity especially when a graph needs to be repetitively reduced. Also, the aforementioned methods rely solely on graph structures but ignore node features. Kumar et al. (2022) propose to

preserve both spectral properties and node features. However, it models the two objectives as separate optimization terms, thus the efficiency problem from the spectral-based methods remains.

## 3 PROBLEM STATEMENT

Our main objective is to construct a continual learning framework on streaming graphs to overcome the catastrophic forgetting problem. Suppose a GNN model is trained for sequential node classification tasks with no access to previous training data, but it can utilize a memory buffer with a limited capacity to store useful information. The goal is to optimize the prediction accuracy of a model on all tasks in the end by minimizing its forgetting of previously acquired knowledge when learning new tasks. In this work, we focus on time-stamped graphs, and the tasks are defined based on the time stamps of nodes in a graph. For each task, the GNN model is trained on a subgraph where source nodes belonging to a specified time period, and all tasks are ordered in time. Node attributes and class labels are only available if the nodes belong to the current time period. The aforementioned setup closely resembles real-life scenarios. We formulate the above problems as follows.

**Problem 1. Continual learning on time-stamped graphs**   We are given a time-stamped expanding graph $\mathcal{G} = (\mathcal{V}, \mathcal{E}, A, X)$, where $\mathcal{V}$ denotes the node set, $\mathcal{E}$ denotes the edge set, $A \in \mathbb{R}^{|\mathcal{V}| \times |\mathcal{V}|}$ and $X \in \mathbb{R}^{|\mathcal{V}| \times d_X}$ denote the adjacency matrix and node features, respectively; Each node $v \in \mathcal{V}$ is assigned to a time period $\tau(v)$. We define a sequence of subgraphs, $\mathcal{G}_1, ..., \mathcal{G}_k$, such that each subgraph $\mathcal{G}_t = (\mathcal{V}_t, \mathcal{E}_t, A_t, X_t)$ from $\mathcal{G}$ based on the following rules:

- For edges in $\mathcal{G}_t$:  $e = (s, o) \in \mathcal{E}_t \Leftrightarrow e \in \mathcal{E}$ and $\tau(s) = t$,
- For nodes in $\mathcal{G}_t$:  $v \in \mathcal{V}_t \Leftrightarrow \tau(v) = t$ or $((s, v) \in \mathcal{E}$ and $\tau(s) = t)$,

where $s$ is a source node and $o$ (or $v$) is a target node. We can assume $\tau(o) \leq \tau(s)$ for $(s, o) \in \mathcal{E}$ (e.g. in a citation network, a paper can not cite another paper published in the future).

Under this setting, we implement a GNN to perform node classification tasks and sequentially train the model on $\mathcal{G}_1, ..., \mathcal{G}_k$. The objective is to optimize the overall performance of the model on all tasks when the model is incrementally trained with new tasks. Note that each taks corresponds to a time period and its subgraph $\mathcal{G}_t$. When the model is trained with a new task $T_t$, it has no access to $\mathcal{G}_1, ..., \mathcal{G}_{t-1}$ and $\mathcal{G}_{t+1}, ..., \mathcal{G}_k$. However, a small memory buffer is allowed to preserve useful information from previous tasks.

**Problem 2. Graph coarsening**   Given a graph $\mathcal{G} = (\mathcal{V}, \mathcal{E})$ with $n = |\mathcal{V}|$ nodes, the goal of graph coarsening is to reduce it to a target size $n'$ with a specific ratio $\gamma$ where $n' = \lfloor \gamma \cdot n \rfloor, 0 < \gamma < 1$. We construct the coarsened graph $\mathcal{G}^r = (\mathcal{V}^r, \mathcal{E}^r)$ through partitioning $\mathcal{V}$ to $n'$ disjoint clusters $(C_1, ..., C_{n'})$, so that each cluster becomes a node in $\mathcal{G}_r$. The construction of these clusters (i.e., the partition of a graph) depends on coarsening strategies. The node partitioning/clustering information can be represented by a matrix $Q \in \mathbb{B}^{n \times n'}$. If we assign every node $i$ in cluster $C_j$ with the same weight, then $Q_{ij} = 1$; If node $i$ is not assigned to cluster $C_j$, $Q_{ij} = 0$. Let $c_j$ be the number of node in $C_j$ and $C = diag(c_1, ..., c'_n)$. The normalized version of $Q$ is $P = QC^{1/2}$. It is easy to prove $P$ has orthogonal columns ($PP^{-1} = I$), and $P_{ij} = 1/\sqrt{c_j}$ if node $i$ belongs to $C_j$; $P_{ij} = 0$ if node $i$ does not belong to $C_j$. The detailed coarsening algorithm will be discussed in the next section.

## 4 METHODOLOGY

We propose **TA$\mathbb{C}\mathbb{O}$** in a continual learning setting to consolidate knowledge learned from proceeding tasks by replaying previous "experiences" to the model. We observe that the majority of experience replay methods, including those tailored for GNN, do not adequately maintain the intricate graph topological properties from previous tasks. Moreover, in a streaming graph setup they fail to capture the inter-dependencies between tasks that result from the presence of overlapping nodes, which can be essential for capturing the dynamic "receptive field" (neighborhood) of nodes and improving the performance on both new and old tasks (Kim et al., 2022). To overcome these limitations, we design a new replay method that preserves both the node attributes and graph topology from previous tasks. Our intuition is that, if we store the original graphs from the old task, minimal old knowledge

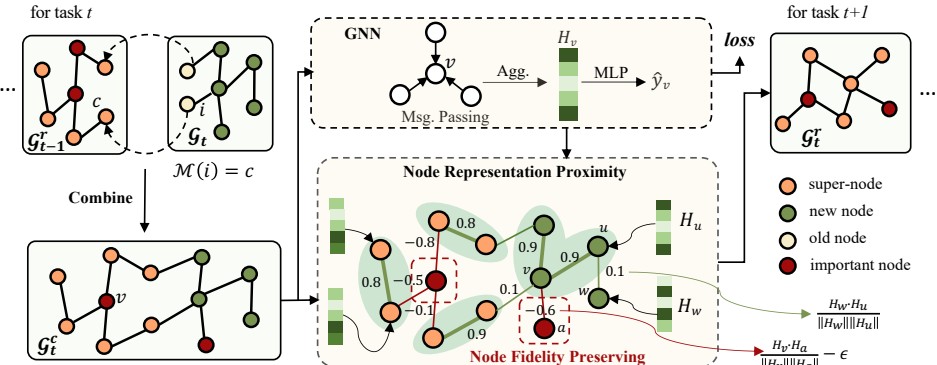

Figure 2: An overview of **TA$\mathbb{CO}$**. At $t$-th time period, the model takes in the coarsened graph $\mathcal{G}_{t-1}^r$ from the last time period and the original graph $\mathcal{G}_t$ from the current time period, and combine them into $\mathcal{G}_t^c$; for the same time period, the selected important node set is updated with the new nodes; the model is then trained on $\mathcal{G}_t^c$ with both the new nodes and the super-nodes from the past; finally $\mathcal{G}_t^c$ is coarsened to $\mathcal{G}_t^r$ for the next time period.

would be lost, but it is also exceedingly inefficient and goes against the initial intention of continual learning. Thus, as an alternative, we coarsen the original graphs to a much smaller size which preserves the important properties (such as node features and graph topologies) of the original graphs. We propose an efficient graph coarsening algorithm based on Node **Re**presentation **Pro**ximity as a key component of **TA$\mathbb{CO}$**. Additionally, we develop a strategy called *Node Fidelity Preservation* for selecting representative nodes to retain high-quality information. An overview of the proposed framework is provided in Figure 2. We provide the pseudo-code of **TA$\mathbb{CO}$** and **RePro** in Appendix A. We also provide the source code in Supplemental Materials.

**Overall framework** We summarize the procedure of our framework as three steps: *combine*, *reduce*, and *generate*. At task $t$, we *combine* the new graph $\mathcal{G}_t$ with the reduced graph $\mathcal{G}_{t-1}^r$ from the last task. Then we *reduce* the combined graph $\mathcal{G}_t^c$ to a set of clusters. At last, we *generate* the contributions of nodes in each cluster to form a super-node in the reduced graph $\mathcal{G}_t^r$. The last step decides the new node features and the adjacency matrix of the reduced graph. We convey the details of each step below.

(1) **Combine:** We use $\mathcal{M}$ (e.g., a hash table) to denote the mapping of each original node to its assigned cluster (super-node) in a reduced graph $\mathcal{G}^r$. In the beginning, we initialize $\mathcal{G}_0^r$ as an empty undirected graph and $\mathcal{M}_0$ as an empty hash table. At task $t$, the model holds copies of $\mathcal{G}_{t-1}^r$, $\mathcal{M}_{t-1}$ and an original graph $\mathcal{G}_t$ for the current task. $\mathcal{G}_t$ contains both new nodes from task $t$ and old nodes that have appeared in previous tasks. We first "combine" $\mathcal{G}_t$ with $\mathcal{G}_{t-1}^r$ to form a new graph $\mathcal{G}_t^c$ by aligning the co-existing nodes in $\mathcal{G}_t$ and $\mathcal{G}_{t-1}^r$ and growing $\mathcal{G}_{t-1}^r$ with the new nodes appeared in $\mathcal{G}_t$. By doing so, we connect the old graph and the new graph, and capture the inter-correlation among tasks. We train the model $f$ to perform node classification tasks on the combined graph $\mathcal{G}_t^c = (A_t^c, X_t^c, Y_t^c)$ with the objective $\arg\min_\theta \ell(f(A_t^c, X_t^c, \theta), Y_t^c)$, where $f$ is a $L$-layer GNN model (e.g. GCN), $\theta$ denotes trainable parameters of the model, and $\ell$ is the loss function. In this work, each new node and old node in $\mathcal{G}_t^c$ contribute equally to the loss during the training process. However, it remains an option to assign distinct weights to these nodes to ensure a balance between learning new information and consolidating old knowledge. We describe a more detailed process in Appendix B.1.

(2) **Reduce:** We decide how nodes are grouped into clusters and each cluster forms a new super-node in the reduced graph. We propose an efficient graph coarsening method, **RePro**, by leveraging the **Re**presentation **Pro**ximities of nodes to reduce the size of a graph through merging "similar" nodes to a super-node. Node representations are automatically learned via GNN models without extra computing processes. We think two nodes are deemed similar based on three factors: 1) *feature similarity*, which evaluates the closeness of two nodes based on their features; 2) *neighbor similarity*, which evaluates two nodes based on their neighborhood characteristics; 3) *geometry closeness*, which

measures the distance between two nodes in a graph (e.g., the length of the shortest path between them). Existing graph coarsening methods concentrate on preserving spectral properties, only taking graph structures into account and disregarding node features. However, estimating spectral similarity between two nodes is typically time-consuming, even with approximation algorithms, making it less scalable for our applications where graphs are dynamically expanded and coarsened. Thus, we aim to develop a more time-efficient algorithm that considers the aforementioned similarity measures.

To get started, we have the combined graph $\mathcal{G}_t^c$ to be coarsened. We train a GNN model with $L$ ($L = 2$ in our case) layers on $\mathcal{G}_t^c$, such that the node embedding of $\mathcal{G}_t^c$ at the first layer (before the activation function) is denoted by

$$H \in \mathbb{R}^{n_t^c \times d^h} = \text{GNN}^{(1)}(A_t^c, X_t^c, \theta), \tag{1}$$

where $d^h$ is the size of the first hidden layer in GNN. The similarity between every connected node pair $(u, v) = e \in \mathcal{E}_t^c$ is calculated based on cosine similarity as $\beta(e) = \frac{H_u \cdot H_v}{\|H_u\|\|H_v\|}$, where $\beta(e)$ is the similarity score between the two end nodes of the edge $e$, $H_i$ is the embedding for node $i$, and $\|\cdot\|$ is the second norm of a vector. We then sort all edges of $\mathcal{G}_t^c$ such that $\beta(e_1) \geq \beta(e_2) \geq ... \geq \beta(e_{m_t^c})$, and we recursively merge (assign the nodes to the same cluster) the two end nodes of the edges that have the largest similarity scores, until the target size is reached.

***Complexity*** The time complexity of such an approach is proportional to the number of levels which can be large, especially when the graph is sparse or the target size is small. In our algorithm, we relax this constraint and allow one node to be merged multiple times continuously. The time complexity of our method is $\mathcal{O}(d^h \cdot m_t^c)$ where $m_t^c$ is the number of edges in the current graph $\mathcal{G}_t^c$.

***Node Fidelity Preservation*** After multiple rounds of coarsening, the quality of a graph deteriorates as its node features and labels are repeatedly processed. Furthermore, the use of a majority vote to determine the label of a cluster can lead to the gradual loss of minority classes and cause a "vanishing minority class" problem.

**Theorem 4.1.** *Consider $n$ nodes with $c$ classes, such that the class distribution of all nodes is represented by $\mathbf{p} = p_1, p_2, ..., p_c$, where $\sum_{i=1}^c p_i = 1$. If these nodes are randomly partitioned into $n'$ clusters such that $n' = \lfloor \gamma \cdot n \rfloor$, $0 < \gamma < 1$ and the class label for each cluster is determined via majority voting. The class distribution of all the clusters is $\mathbf{p}' = p_1', p_2', ..., p_c'$ where $\mathbf{p}_i'$ is computed as the ratio of clusters labeled as class $i$ and $\sum_{i=1}^c p_i' = 1$. Let $k$ be one of the classes, and the rest of the class are balanced $p_1 = ... = p_{k-1} = p_{k+1} = ... = p_c$. It holds that:*

*1. If $p_k = 1/c$ and all classes are balanced $p_1 = p_2 = ... = p_c$, then $\mathbb{E}[p_k'] = p_k$.*

*2. When $p_k < 1/c$, $\mathbb{E}[p_k'] < p_k$, and $\mathbb{E}[\frac{p_k'}{p_k}]$ decreases as $n'$ decreases. There exists a $p^{min}$ such that $0 < p^{min} < 1$, and when $p_k < p^{min}$, $\mathbb{E}[\frac{p_k'}{p_k}]$ decrease as $p_k$ decreases.*

The proof of Theorem 4.1 is provided in Appendix B.2. Theorem 4.1 shows that as the ratio of a class decreases, its decline becomes more severe when the graph is reduced. Eventually, the class may even disappear entirely from the resulting graph. To combat these issues, we suggest preserving representative nodes in a "replay buffer" denoted as $\mathcal{V}_t^{rb}$. We adopt three strategies from Chaudhry et al. (2019) to select representative nodes, namely *Reservoir Sampling*, *Ring Buffer*, and *Mean of Features*. The replay buffer has a fixed capacity and is updated as new nodes are added. During the coarsening process, we prevent the selected nodes in $\mathcal{V}_t^{rb}$ from being merged by imposing a penalty on the similarity score $\beta(e)$ such that

$$\beta(e) = \frac{H_u \cdot H_v}{\|H_u\|\|H_v\|} - \mathbb{1}(u \in \mathcal{V}_t^{rb} \text{ or } v \in \mathcal{V}_t^{rb}) \cdot \epsilon, \tag{2}$$

where $\epsilon$ is a constant and $\epsilon > 0$. It is worth noting that we do not remove the edges of these nodes from the list of candidates. Instead, we assign a high value to the penalty $\epsilon$. This is to prevent scenarios where these nodes play a critical role in connecting other nodes. Removing these nodes entirely may lead to the graph not being reduced to the desired size due to the elimination of important paths passing through them. We make the following observation:

**Observation 4.1.** *Node Fidelity Preservation with buffer size $b$ can alleviate the declination of a minority class $k$ when $p_k$ decreases and $n'$ decreases, and prevent class $k$ from vanishing when $p_k$ is small.*

See Appendix B.2 for further discussions.

Note that **RePro** does not require any additional parameters or training time despite relying on the learned node embeddings. We train a GNN model on a combined graph at each time step for node classification tasks. The node embeddings learned from the GNN model at different layers are representative of the nodes' neighborhoods. We use this fact and propose to measure the similarity of two nodes based on the distance between their embedding vectors. However, it takes quadratic time to calculate the pair-wise distance among nodes, thus we make a constraint that only connected nodes can be merged. Since connectivity has also been used to estimate the geometry closeness of two nodes (Pan et al., 2010), by doing so, we are able to cover the three similarity measures as well as reduce the time complexity to linear time in terms of the number of edges to calculate node similarities.

Our proposed approach based on node representations seems to be distinct from spectral-based methods, but they share a similar core in terms of the preserving of graph spectral properties. See Appendix B.3 for more details.

(3) **Generate:** From the last step, we get the membership matrix $Q_t \in \mathbb{B}^{n_t^c \times n_t^r}$ where $n_t^c$ denotes the number of nodes in the combined graph, $n_t^r = \lfloor \gamma \cdot n_t^c \rfloor$ denotes the number of nodes in the coarsened graph and $\gamma$ is the coarsening ratio. $Q_t[i,j] = 1$ denotes that node $i$ is assigned to super-node $j$. Otherwise, $Q_t[i,j] = 0$.

A simple way to normalize $Q_t$ is assuming each node contributes equally to their corresponding super-node (e.g. $Q_t[i,j] = 1/\sqrt{c_j}$ for all any node $i$ that belongs to cluster/supernode $j$). However, nodes might have varying contributions to a cluster depending on their significance. Intuitively, when a node is identified as a very popular one that is directly or indirectly connected with a large number of other nodes, preserving more of its attributes can potentially mitigate the effects of the inevitable "information degrading" caused by the graph coarsening procedure. To address the above issue, we propose to use two different measures to decide a node's importance score: 1) *node degree*: the number of 1-hop neighbors of the node. 2) *neighbor degree sum*: the sum of the degrees of a node's 1-hop neighbors. In this step, we propose to normalize $Q_t$ to $P_t$ utilizing these node importance information. We calculate the member contribution matrix $P_t \in \mathbf{R}^{n_t^c \times n_t^r}$. Let $i$ be a node belonging to cluster $C_j$ at timestamp $t$, and $s_i > 0$ be the importance score of node $i$ (node degree or neighbor degree sum), then $p_{t,(ij)} = \sqrt{\frac{s_i}{\sum_{v \in C_j} s_v}}$. It is straightforward to prove that $P_t^\top P_t = I$ still holds.

Once we have $P_t$, we get the new reduced graph $\mathcal{G}_t^r = (A_t^r, X_t^r, Y_t^r)$ as:

$$A_t^r = Q_t^\top A_t^c Q_t, \quad X_t^r = P_t^\top X_t^c, \quad Y_t^r = \arg\max(P_t^\top Y_t^c), \tag{3}$$

where the label for each cluster is decided by a majority vote. Only partial nodes are labeled, and the rows of $Y_t^c$ for those unlabelled nodes are zeros and thus do not contribute to the vote.

Through training all tasks, the number of nodes in the reduced graph $\mathcal{G}^r$ is upper-bounded by $\frac{1-\gamma}{\gamma} \cdot (n_{\text{MAX}})$, where $n_{\text{MAX}}$ is the largest number of the new nodes for each task (See Appendix B.4 for proof); when the reduction ratio $\gamma$ is 0.5, the expression above is equivalent to $n_{\text{MAX}}$, meaning the size of the reduced graph is roughly the same size with the original graph for each task. The number of edges $m$ is bounded by $n_{\text{MAX}}^2$, but we observe generally $m \ll n_{\text{MAX}}^2$ in practice.

## 5 EMPIRICAL EVALUATION

We conduct experiments on time-stamped graph datasets: Kindle (He & McAuley, 2016; McAuley et al., 2015), DBLP (Tang et al., 2008) and ACM (Tang et al., 2008) to evaluate the performance of **TA**ℂ𝕆. See Appendix C.1 for the details of the datasets and C.2 for hyperparameter setup.

**Comparison methods** We compare the performance of **TA**ℂ𝕆 with SOTA continual learning methods including EWC (Kirkpatrick et al., 2017), GEM (Lopez-Paz & Ranzato, 2017), TWP (Liu et al., 2020), OTG (Feng et al., 2023), ERGNN (Zhou et al., 2020), SSM (Zhang et al., 2022a), DyGrain (Kim et al., 2022), IncreGNN (Wei et al., 2022), and SSRM (Su & Wu, 2023). EWC and GEM were previously not designed for graphs, so we train a GNN on new tasks but ignore the graph structure when applying continual learning strategies. ERGNN-rs, ERGNN-rb, and ERGNN-mf are ERGNN methods with different memory buffer updating strategies: Reservoir Sampling (rs), Ring Buffer (rb), and Mean of Features (mf) (Chaudhry et al., 2019). SSRM is an additional regularizer

Table 1: Node classification performance with GCN as the backbone on three datasets (averaged over 10 trials). Standard deviation is denoted after $\pm$.

| Method | Kindle | | DBLP | | ACM | |
|---|---|---|---|---|---|---|
| | F1-AP(%) | F1-AF (%) | F1-AP (%) | F1-AF (%) | F1-AP (%) | F1-AF (%) |
| joint train | 87.21 $\pm$ 0.55 | 0.45 $\pm$ 0.25 | 86.33 $\pm$ 1.38 | 0.77 $\pm$ 0.13 | 75.35 $\pm$ 1.49 | 1.87 $\pm$ 0.60 |
| finetune | 69.10 $\pm$ 10.85 | 18.99 $\pm$ 11.19 | 67.85 $\pm$ 8.05 | 20.43 $\pm$ 7.07 | 60.53 $\pm$ 9.35 | 19.09 $\pm$ 9.23 |
| simple-reg | 68.80 $\pm$ 10.02 | 18.21 $\pm$ 10.49 | 69.70 $\pm$ 9.16 | 18.69 $\pm$ 8.48 | 61.63 $\pm$ 10.09 | 17.83 $\pm$ 9.99 |
| EWC | 77.08 $\pm$ 8.37 | 10.87 $\pm$ 8.62 | 79.38 $\pm$ 4.86 | 8.85 $\pm$ 4.11 | 66.48 $\pm$ 6.43 | 12.73 $\pm$ 6.26 |
| TWP | 78.90 $\pm$ 4.71 | 8.99 $\pm$ 4.93 | 80.05 $\pm$ 3.71 | 8.23 $\pm$ 3.28 | 65.98 $\pm$ 7.26 | 13.33 $\pm$ 6.94 |
| OTG | 69.01 $\pm$ 10.55 | 18.94 $\pm$ 10.79 | 68.24 $\pm$ 10.12 | 20.12 $\pm$ 9.34 | 61.45 $\pm$ 9.94 | 18.33 $\pm$ 9.86 |
| GEM | 76.08 $\pm$ 6.70 | 11.01 $\pm$ 7.27 | 80.04 $\pm$ 3.24 | 7.90 $\pm$ 2.68 | 67.17 $\pm$ 4.24 | 11.69 $\pm$ 3.94 |
| ERGNN-rs | 77.63 $\pm$ 3.61 | 9.64 $\pm$ 4.19 | 78.02 $\pm$ 5.79 | 10.08 $\pm$ 5.16 | 64.82 $\pm$ 7.89 | 14.43 $\pm$ 7.68 |
| ERGNN-rb | 75.87 $\pm$ 6.41 | 11.46 $\pm$ 6.98 | 75.16 $\pm$ 7.24 | 12.85 $\pm$ 6.54 | 63.58 $\pm$ 8.82 | 15.66 $\pm$ 8.71 |
| ERGNN-mf | 77.28 $\pm$ 5.91 | 10.15 $\pm$ 6.31 | 77.42 $\pm$ 5.25 | 10.64 $\pm$ 4.38 | 64.80 $\pm$ 8.49 | 14.59 $\pm$ 8.41 |
| DyGrain | 69.14 $\pm$ 10.47 | 18.88 $\pm$ 10.72 | 67.52 $\pm$ 10.88 | 20.83 $\pm$ 10.16 | 61.40 $\pm$ 9.57 | 18.47 $\pm$ 9.50 |
| IncreGNN | 69.45 $\pm$ 10.34 | 18.48 $\pm$ 10.66 | 69.40 $\pm$ 9.60 | 18.92 $\pm$ 8.75 | 61.32 $\pm$ 9.70 | 18.42 $\pm$ 9.64 |
| SSM | 78.99 $\pm$ 3.13 | 8.19 $\pm$ 3.63 | 82.71 $\pm$ 1.76 | 4.20 $\pm$ 1.26 | 68.77 $\pm$ 2.93 | 9.50 $\pm$ 2.47 |
| SSRM | 77.37 $\pm$ 4.06 | 9.99 $\pm$ 4.55 | 77.43 $\pm$ 5.34 | 10.66 $\pm$ 4.47 | 64.39 $\pm$ 7.43 | 14.72 $\pm$ 7.48 |
| **TA$\mathbb{C}\mathbb{O}$** | **82.97** $\pm$ **2.05** | **4.91** $\pm$ **1.90** | **84.60** $\pm$ **2.01** | **2.51** $\pm$ **1.03** | **70.96** $\pm$ **2.68** | **8.02** $\pm$ **2.33** |
| p-value | <0.0001 | <0.0001 | 0.002 | <0.0001 | 0.005 | 0.02 |

to be applied on top of a CGL framework; we choose ERGNN-rs as the base CGL model. Besides, finetune provides the estimated lower bound without any strategies applied to address forgetting problems, and joint-train provides an empirical upper bound where the model has access to all previous data during the training process.

We also compare **RePro** with five representative graph coarsening SOTA methods. We replace the coarsening algorithm in **TA$\mathbb{C}\mathbb{O}$** with different coarsening algorithms. Alge. JC (Chen & Safro, 2011), Aff.GS (Livne & Brandt, 2012), Var. edges (Loukas, 2018), and Var. neigh (Loukas, 2018) are graph spectral-based methods; FGC (Kumar et al., 2022) consider both graph spectrals and node features. We follow the implementation of Huang et al. (2021) for the first four coarsening methods.

**Evaluation metrics** We use *Average Performance* (AP↑) and *Average Forgetting* (AF↓) (Chaudhry et al., 2019) to evaluate the performance on test sets. AP and AF are defined as AP $= \frac{1}{T} \sum_{j=1}^{T} a_{T,j}$, AF $= \frac{1}{T} \sum_{j=1}^{T} \max_{l \in \{1,...,T\}} a_{l,j} - a_{T,j}$, where T is the total number of tasks and $a_{i,j}$ is the prediction metric of the model on the test set of task $j$ after it is trained on task $i$. The prediction performance can be measured with different metrics. In this paper, we use macro F1 and balanced accuracy score (BACC). F1-AP and F1-AF indicate the AP and the AF for macro F1 and likewise for BACC-AP and BACC-AF. We follow Grandini et al. (2020) to calculate the macro F1 and BACC scores for multi-class classification.

**Main results** We evaluate the performance of **TA$\mathbb{C}\mathbb{O}$** and other baselines on three datasets and three backbone GNN models, including GCN (Kipf & Welling, 2016), GAT (Veličković et al., 2017), and GIN (Xu et al., 2019). We only report the node classification performance in terms of F1-AP and F1-AF with GCN in Table 1 due to the space limit. See Appendix D.1 for complete results. We report the average values and the standard deviations over 10 runs. It shows that **TA$\mathbb{C}\mathbb{O}$** outperforms the best state-of-the-art CGL baseline method with high statistical significance, as evidenced by p-values below 0.05 reported from a t-test. Additionally, we note that despite being Experience Replay-based methods, ER-rs, ER-rb, and ER-mf do not perform as well as SSM and **TA$\mathbb{C}\mathbb{O}$**, highlighting the importance of retaining graph structural information when replaying experience nodes to the model. Furthermore, we infer that **TA$\mathbb{C}\mathbb{O}$** outperforms SSM due to its superior ability to preserve graph topology information and capture task correlations through co-existing nodes.

**Graph coarsening methods** We evaluate the performance of **RePro** by replacing the graph coarsening module of **TA$\mathbb{C}\mathbb{O}$** with five widely used coarsening algorithms, while keeping all other components unchanged. The overall results on GCN are presented in Table 2. See Appendix D.2 for complete results. We report the average time in seconds consumed for each model to coarsen the

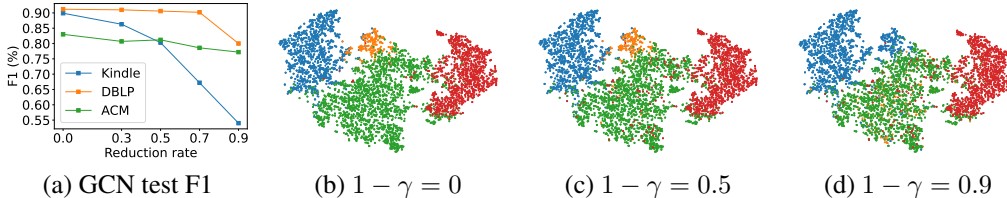

(a) GCN test F1      (b) $1 - \gamma = 0$      (c) $1 - \gamma = 0.5$      (d) $1 - \gamma = 0.9$

Figure 3: (a) The test macro-F1 scores of the GCN model trained on the coarsened graphs with different reduction rates on three datasets. (b)-(d) t-SNE visualization of node embeddings of the DBLP test graph with a reduction rate of 0, 0.5, and 0.9 on the training graph respectively.

graph for each task. The results demonstrate that **RePro** achieves better or comparable prediction performance compared with the best baseline while being considerably more efficient in computing time compared to all other models.

**Graph reduction rates** We examine how **RePro** preserves original graph information. Following the setting in Huang et al. (2021), we train GNNs from scratch on original and coarsened graphs, then compare their prediction performance on the test nodes from the original graph. Using subgraphs from the first task of three datasets as original graphs, we train GCN models on coarsened graphs with different coarsening rates $1 - \gamma$. Figure 3 (a) shows that prediction performance is relatively stable as graphs are reduced for DBLP and ACM, but F1 scores are more sensitive to the reduction rate on Kindle. This may

Table 2: Coarsen runtime (seconds) and node classification results of **TA**ℂ𝕆 variations with different coarsening methods on three datasets with GCN (average over 10 trials). Boldface and blue color indicate the best and the second-best result of each column.

| Method | Kindle | | DBLP | | ACM | |
|---|---|---|---|---|---|---|
| | Time | F1-AP (%) | Time | F1-AP (%) | Time | F1-AP (%) |
| Alge. JC | 8.9 | 81.09±2.15 | 70.8 | **85.24±1.55** | 11.8 | 70.25±3.23 |
| Aff. GS | 65.6 | 77.42±4.44 | 237.1 | 84.34±1.70 | 96.1 | 68.12±5.69 |
| Var. neigh | 6.9 | 80.77±3.66 | 7.3 | 84.91±1.38 | 10.3 | 66.83±7.53 |
| Var. edges | 10.1 | 82.29±1.95 | 28.0 | 85.13±1.86 | 13.8 | **71.42±2.48** |
| FGC | 7.7 | 81.42±2.98 | 10.8 | 84.77±1.69 | 7.0 | 66.97±7.44 |
| **RePro** | **2.3** | **82.97±2.05** | **1.1** | 84.60±2.01 | **1.4** | 70.96±2.68 |

be due to overfitting on smaller datasets. Although reduced graphs may not preserve all information, they can still be used to consolidate learned knowledge and reduce forgetting in CGL paradigm. We also test if the model learns similar node embeddings on coarsened and original graphs. In Figure 3 (b)-(d), we visualize test node embeddings on the DBLP dataset for reduction rates of 0, 0.5, and 0.9 using t-SNE. We observe similar patterns for 0 and 0.5 reduction rates, and gradual changes as the reduction rate increases.

**Additional ablation studies** We also study the performances of CGL models after training on each tasks E.1, the effectiveness of the Node Fidelity Preservation E.5, and the effects of different important node selection strategies E.6. We present those results in Appendix E for those readers who are interested.

## 6 CONCLUSION

In this paper, we present a novel CGL framework, **TA**ℂ𝕆, which stores useful information from previous tasks with a dynamically reduced graph to consolidate learned knowledge. Additionally, we propose an efficient embedding proximity-based graph coarsening method, **RePro**, that can preserve important graph properties. We present a Node Fidelity Preservation strategy and theoretically prove its capability in preventing the vanishing minority class problem. We demonstrate the effectiveness of **TA**ℂ𝕆 and **RePro** on real-world datasets with a realistic streaming graph setup.

**Limitations:** This work only considers situations where nodes and edges are added to streaming graphs with a single relation type. In the future, we plan to investigate continual graph learning methods when nodes and edges can be deleted or modified. Moreover, we will generalize our method to complex graphs such as multi-relation graphs and broader graph tasks such as link prediction for recommendation systems.

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

APPENDIX

## A PSEUDOCODE

The pseudocode of the proposed CGL framework **TA**$\mathbb{CO}$ is described in Algorithm 1, and the proposed graph coarsening module **RePro** is described in Algorithm 2.

---

**Algorithm 1** The Proposed Method **TA**$\mathbb{CO}$

---

1: **Input**: A sequence of graphs $\mathcal{G}_1, ..., \mathcal{G}_k$ ($\mathcal{G}_t$ is only accessible at task $t$)
2: **Output**: A trained GNN node classifier $f$ with parameter $\theta$
3: $\mathcal{G}_0^r, \mathcal{M}_0, \mathcal{V}_0^{rb} \leftarrow \varnothing, \{\}, \{\}$
4: **for** task t = 1 **to** k **do**
5:     $(\mathcal{G}_t^c, \mathcal{M}_{t-1}) \leftarrow \mathsf{combine}(\mathcal{G}_t, \mathcal{G}_{t-1}^r, \mathcal{M}_{t-1})$
6:     **for** epoch=0 **to** num_epoch **do**
7:         Train $f$ on $\mathcal{G}_t^c$ and update $\theta$
8:     **end for**
9:     $\mathcal{V}_t^{rb} \leftarrow \mathsf{SamplingStrategy}(\mathcal{V}_{t-1}^{rb}, \mathcal{G}_t)$
10:    Compute node embedding $H_t$ based on Eq. 1
11:    $P_t, Q_t, \mathcal{P}(\cdot) \leftarrow \mathsf{GraphCoarseningAlgorithm}\ (\mathcal{G}_t^r, H_t, \mathcal{V}_t^{rb})$
12:    Compute $\mathcal{G}_t^r$ based on Eq. 3.
13:    $\mathcal{M}_t \leftarrow \mathcal{M}_t(v) = \mathcal{P}(\mathcal{M}_{t-1}(v))$ for $v \in \mathcal{M}_{t-1}$
14: **end for**
15:
16: **function** COMBINE($\mathcal{G}_t, \mathcal{G}_{t-1}^r, \mathcal{M}_{t-1}$)
17:     $\mathcal{G}_t^c \leftarrow \mathcal{G}_{t-1}^r$
18:     **for** each edge $(s, o) \in \mathcal{E}_t$ **do**
19:         **if** $\mathcal{T}(s) = \mathcal{T}(o) = t$ **then**
20:             Add $s$ and $o$ to $\mathcal{G}_t^c$ and $\mathcal{M}_{t-1}$
21:             Add an undirected edge from $s$ to $o$ on $\mathcal{G}_t^c$
22:         **else**
23:            **if** $\mathcal{T}(s) = t$ **and** $\mathcal{T}(s) < t$ **and** $o \in \mathcal{M}_{t-1}$ **then**
24:            Add $s$ to $\mathcal{G}_t^c$ and $\mathcal{M}_{t-1}$
25:            Add an undirected edge from $s$ to $\mathcal{M}_{t-1}(o)$ on $\mathcal{G}_t^c$
26:         **end if end if**
27:     **end for**
28:     **Return** $\mathcal{G}_t^c, \mathcal{M}_{t-1}$
29: **end function**

---

## B SUPPLEMENTAL METHODOLOGY

### B.1 OVERALL FRAMEWORK

In the proposed framework, during the training process, we use a reduced graph $\mathcal{G}^r$ as an approximate representation of previous graphs, and a function $\mathcal{M}(\cdot)$ (e.g., a hash table) to map each original node to its assigned cluster (super-node) in $\mathcal{G}^r$ (e.g., if an original node $i$ is assigned to cluster $j$, then $\mathcal{M}(i) = j$). Both $\mathcal{G}_t^r$ and $\mathcal{M}_t(\cdot)$ are updated after the $t$-th task.

To get started we initialize $\mathcal{G}_0^r$ as an empty undirected graph and $\mathcal{M}_0$ as an empty hash table. At task $T_t$, the model holds copies of $\mathcal{G}_{t-1}^r$, $\mathcal{M}_{t-1}$ and an original graph $\mathcal{G}_t$ for the current task.

We combine $\mathcal{G}_t$ with $\mathcal{G}_{t-1}^r$ to form a new graph $\mathcal{G}_t^c$ according to the following procedure:

1. Initialize the combined graph $\mathcal{G}_t^c = (A_t^c, X_t^c, Y_t^c)$ as $\mathcal{G}_{t-1}^r$ such that $A_t^c = A_{t-1}^r$, $X_t^c = X_{t-1}^r$, and $Y_t^c = Y_{t-1}^r$;
2. *Case 1: new source node and new target node.* For each edge $(s, o) \in \mathcal{E}_t$, if $\tau(s) = \tau(o) = t$, we add $s$ and $o$ to $\mathcal{G}_t^c$, and we add an undirected edge $(s,o)$ to $\mathcal{G}_t^c$ ;

---

**Algorithm 2** The Proposed Graph Coarsening Algorithm **RePro**

---

1: **Input**: The original graph $\mathcal{G}$, node embedding matrix $H$, node sets $V^{rb}$, and the reduction rate $\gamma$
2: **Output**: partition matrix $Q$, normalized partition matrix $P$, and the mapping function $\mathcal{P}(\cdot)$
3: Initialize the mapping function $\mathcal{P}(\cdot)$ such that $\mathcal{P}(v) = v$ for $v \in \mathcal{V}$
4: $n^r \leftarrow |\mathcal{V}|$
5: $n^{\text{target}} \leftarrow \lfloor r \cdot |\mathcal{V}| \rfloor$
6: Sort all edges $e \in \mathcal{E}$ in the descending order based on their similarity scores calculated according to Eq. 2
7: **for** each edge $e = (u, v) \in \mathcal{E}$ **do**
8:     **if** $\mathcal{P}(u) \neq \mathcal{P}(v)$ **then**
9:         // if $u$ and $v$ are in different clusters
10:         Merge the clusters of $u$ and $v$ such that $\mathcal{P}(u) = \mathcal{P}(v)$
11:         $n^r = n^r - 1$
12:     **end if**
13:     **if** $n^r \leq n^{\text{target}}$ **then**
14:         Break
15:     **end if**
16: **end for**
17: Construct $Q$ with $\mathcal{P}(\cdot)$; compute $P$ based on $p_{t,(ij)} = \sqrt{\frac{s_i}{\sum_{v \in C_j} s_v}}$.
18: Return $Q$, $P$, $\mathcal{P}(\cdot)$

---

3. *Case 2: new source node and old target node.* If $\tau(s) = t$, but $\tau(o) < t$, and $o \in \mathcal{M}_{t-1}$, we add $s$ to $\mathcal{G}_t^c$, and we add an undirected edge $(s, \mathcal{M}_{t-1}(o))$ to $\mathcal{G}_t^c$;

4. *Case 3: deleted target node.* If $\tau(o) < t$ and $o \notin \mathcal{M}_{t-1}$, we ignore the edge.

5. When a new node $v$ is added to $\mathcal{G}_{t-1}^c$, it is also added to $\mathcal{M}_{t-1}$ and is assigned to a new cluster.

It is worth noting that we use directed edges to better explain the above procedure. In our implementation, the combined graph is undirected since the directness of the edges of the combined graph is not critical in learning node embeddings in a graph like a citation network.

## B.2 NODE FIDELITY PRESERVATION

**Theorem 4.1.** *Consider $n$ nodes with $c$ classes, such that the class distribution of all nodes is represented by $\mathbf{p} = p_1, p_2, ..., p_c$, where $\sum_{i=1}^{c} p_i = 1$. If these nodes are randomly partitioned into $n'$ clusters such that $n' = \lfloor \gamma \cdot n \rfloor$, $0 < \gamma < 1$ and the class label for each cluster is determined via majority voting. The class distribution of all the clusters is $\mathbf{p}' = p_1', p_2', ..., p_c'$ where $p_i'$ is computed as the ratio of clusters labeled as class $i$ and $\sum_{i=1}^{c} p_i' = 1$. Let $k$ be one of the classes, and the rest of the class are balanced $p_1 = ... = p_{k-1} = p_{k+1} = ... = p_c$. It holds that:*

*1. If $p_k = 1/c$ and all classes are balanced $p_1 = p_2 = ... = p_c$, then $\mathbb{E}[p_k'] = p_k$.*

*2. When $p_k < 1/c$, $\mathbb{E}[p_k'] < p_k$, and $\mathbb{E}[\frac{p_k'}{p_k}]$ decreases as $n'$ decreases. There exists a $p^{min}$ such that $0 < p^{min} < 1$, and when $p_k < p^{min}$, $\mathbb{E}[\frac{p_k'}{p_k}]$ decrease as $p_k$ decreases.*

*Proof.* We prove the theorem by deriving the value of $\mathbb{E}[p_k']$. Since $\mathbb{E}[p_k']$ is invariant to the order of the classes, for convenience, we consider $i = 1$ without losing generality. The probability of the first class after the partitioning is:

$$\mathbb{E}[p_1'] = \frac{1}{n'} \sum_{a=1}^{n-n'} \mathbb{E}[n_a] q(a, \mathbf{p}), \tag{4}$$

where $\mathbb{E}[n_a]$ is the expectation of the number of clusters containing $a$ nodes, and $q(a, \mathbf{p})$ is the probability that class 1 is the majority class in a cluster with size $a$.

**Property B.1.** *The expectation of the number of clusters containing a node is*

$$\mathbb{E}[n_a] = n' \times \binom{n - n' + 1}{a - 1} \left(\frac{1}{n'}\right)^{a-1} \left(1 - \frac{1}{n'}\right)^{n-n'-(a-1)}.$$

*Proof.* Define $I_j$ to be the indicator random variable for the $j^{th}$ cluster, such that:

$$I_j = \begin{cases} 1 & \text{if the } j^{th} \text{ cluster contains exactly } a \text{ samples,} \\ 0 & \text{otherwise.} \end{cases}$$

We first compute the expected value of $I_j$ for a single cluster. Let's calculate the probability that $a - 1$ out of the $n - n'$ samples are allocated to the $j^{th}$ cluster:

(a) There are $\binom{n-n'}{a-1}$ ways to choose $a - 1$ samples from the $n - n'$ remaining samples.

(b) The probability that each of these $a - 1$ samples is placed in the $j^{th}$ cluster is $\left(\frac{1}{n'}\right)^{a-1}$.

(c) The remaining $n - n' - a + 1$ samples from our original pool of $n - n'$ should not be allocated to the $j^{th}$ cluster. The probability that all of them avoid this cluster is $\left(1 - \frac{1}{n'}\right)^{n-n'-(a-1)}$.

Thus, the probability that the $j^{th}$ cluster contains exactly $x$ samples is:

$$\mathbb{E}[I_j] = \binom{n - n'}{a - 1} \left(\frac{1}{n'}\right)^{a-1} \left(1 - \frac{1}{n'}\right)^{n-n'-(a-1)}. \tag{5}$$

The expected number of clusters out of all $n'$ clusters that contain exactly $a$ samples is:

$$\mathbb{E}[n_a] = \sum_{j=1}^{n'} \mathbb{E}[I_j]. \tag{6}$$

Given that each $I_j$ is identically distributed, we can simplify the sum:

$$\mathbb{E}[n_a] = n' \times \mathbb{E}[I_j]. \tag{7}$$

Substituting the expression derived for $\mathbb{E}[I_j]$ from step 2, we obtain:

$$\mathbb{E}[n_a] = n' \times \binom{n - n'}{a - 1} \left(\frac{1}{n'}\right)^{a-1} \left(1 - \frac{1}{n'}\right)^{n-n'-(a-1)}. \tag{8}$$

$\square$

It is easy to show that when $p_1 = p_2 = ... = p_c$, it holds that $q(a, \mathbf{p}) = \frac{1}{c}$ since all classes are equivalent and have equal chances to be the majority class. Thus:

$$\mathbb{E}[p_1'] = \frac{1}{n'} \sum_{a=1}^{n'} \mathbb{E}[n_a] q(a, \mathbf{p}) \tag{9}$$

$$= \frac{1}{n'} \sum_{a=1}^{n'} \mathbb{E}[n_a] \frac{1}{c} \tag{10}$$

$$= \frac{1}{n'} \sum_{a=1}^{n'} n' \times \binom{n - n'}{a - 1} \left(\frac{1}{n'}\right)^{a-1} \left(1 - \frac{1}{n'}\right)^{n-n'-(a-1)} \frac{1}{c} \tag{11}$$

$$= \frac{1}{n'} n' \frac{1}{c} \tag{12}$$

$$= \frac{1}{c}. \tag{13}$$

To compute $q(a, \mathbf{p})$, we need to consider the situations in which class 1 is the exclusive majority class and the cases where class 1 ties with one or more other classes for having the most samples. In the second case, we roll a die to select the majority class from all the classes that have most samples.

To start, we consider the situation when class 1 has the most nodes and no other classes tie with it. We enumerate all possible combinations of class assignments and calculate the probability of each.

$$q_0(a, \mathbf{p}) = \sum_{i_1=1}^{a} \sum_{i_2=0}^{i_1-1} \cdots \sum_{i_c=0}^{i_1-1} \mathbb{1}\{\sum_{k=1}^{c} i_k = a\} f(\mathbf{i}, a, \mathbf{p}), \tag{14}$$

where $\mathbf{i} = i_1, i_2 \ldots i_c$, and

$$f(\mathbf{i}, a, \mathbf{p}) = \prod_{k=1}^{c} \binom{a - i_1 \cdots - i_k}{i_k} p_k^{i_k} (1 - p_k)^{a - i_1 - \cdots - i_k}, \tag{15}$$

and

$$p_k = \begin{cases} p_1 & \text{if } k = 1 \\ \frac{1-p_1}{c-1} & \text{otherwise} \end{cases}. \tag{16}$$

We then consider the situation when class 1 ties with another class. We first select another class that ties in with class 1, and we enumerate the possibility of other classes. We multiply the whole term by $\frac{1}{2}$ as there is an equal chance to select either class as the majority class.

$$q_1(a, \mathbf{p}) = \frac{1}{2} \sum_{j_1=2}^{c} \left( \sum_{i_1=1}^{a} \sum_{i_2=0}^{i_1-1} \cdots \sum_{i_{j_1}=i_1}^{i_1} \cdots \sum_{i_c=0}^{i_1-1} \mathbb{1}\{\sum_{k=1}^{c} i_k = a\} f(\mathbf{i}, a, \mathbf{p}) \right). \tag{17}$$

We extend the above equation to a general case where class 1 ties with k classes ($1 \leq k \leq c - 1$). Here we need to select k classes that tie with class 1. Class 1 now shares a $\frac{1}{k+1}$ chance to be selected as the majority class with the selected $k$ classes.

$$q_k(a, \mathbf{p}) = \frac{1}{k} \sum_{j_1=2}^{c-k+1} \sum_{j_2=j_1}^{c-k+2} \cdots \sum_{j_k=j_{k-1}}^{c} \left( \sum_{i_1=1}^{a} \sum_{i_2=0}^{i_1-1} \cdots \sum_{i_{j_1}=i_1}^{i_1} \cdots \sum_{i_{j_k}=i_1}^{i_1} \cdots \sum_{i_c=0}^{i_1-1} \mathbb{1}\{\sum_{k=1}^{c} i_k = a\} f(\mathbf{i}, a, \mathbf{p}) \right). \tag{18}$$

Finally, we combine all the cases, and the probability that class 1 is the majority class is:

$$q(a, \mathbf{p}) = \sum_{k=0}^{c-1} q_k(a, \mathbf{p}). \tag{19}$$

The expectation of $\frac{p_1'}{p_1}$ is thus:

$$\mathbb{E}[\frac{p_1'}{p_1}] = \frac{1}{p_1} \frac{1}{n'} \sum_{a=1}^{n'} \mathbb{E}[n_a] q(a, \mathbf{p}). \tag{20}$$

To study the behavior of $\mathbb{E}[\frac{p_1'}{p_1}]$ when $p_1$ changes, we derive the following derivative:

$$\frac{d\mathbb{E}[\frac{p_1'}{p_1}]}{dp_1} = \frac{d\frac{1}{p_1} \frac{1}{n'} \sum_{a=1}^{n'} \mathbb{E}[n_a] q(a, \mathbf{p})}{dp_1} \tag{21}$$

$$= \frac{1}{n'} \sum_{a=1}^{n'} \mathbb{E}[n_a] \frac{d\frac{q(a,\mathbf{p})}{p_1}}{dp_1}, \tag{22}$$

where

$$\frac{d\frac{q(a,\mathbf{p})}{p_1}}{dp_1} = \sum_{k=0}^{c-1} \frac{d\frac{q_k(a,p)}{p_1}}{dp_1} \tag{23}$$

$$= \frac{1}{2} \sum_{j_1=2}^{c-k+1} \sum_{j_2=j_1}^{c-k+2} \cdots \sum_{j_k=j_{k-1}}^{c} \left( \sum_{i_1=1}^{a} \sum_{i_2=0}^{i_1-1} \cdots \sum_{i_{j_1}=i_1}^{i_1} \cdots \sum_{i_{j_k}=i_1}^{i_1} \cdots \sum_{i_c=0}^{i_1-1} \mathbb{1}\{\sum_{k=1}^{c} i_k = a\} \frac{d\frac{f(\mathbf{i},a,\mathbf{p})}{p_1}}{dp_1} \right). \tag{24}$$

To find $\frac{d\frac{f(\mathbf{i},a,\mathbf{p})}{p_1}}{dp_1}$, we first separate the terms that are independent of $p_1$:

$$\frac{f(\mathbf{i},a,\mathbf{p})}{p_1} = \frac{1}{p_1} \prod_{k=1}^{c} \binom{a-i_1\ldots-i_k}{i_k} p_k^{i_k}(1-p_k)^{a-i_1-\ldots-i_k} \tag{25}$$

$$= \frac{1}{p_1} \left( \prod_{k=1}^{c} \binom{a-i_1\ldots-i_k}{i_k} \right) \prod_{k=1}^{c} p_k^{i_k}(1-p_k)^{a-i_1-\ldots-i_k} \tag{26}$$

$$= \left( \prod_{k=1}^{c} \binom{a-i_1\ldots-i_k}{i_k} \right) p_1^{i_1-1}(1-p_1)^{a-i_1}(\frac{1-p_1}{c-1})^{\sum_{k=2}^{c} i_k}(1-\frac{1-p_1}{c-1})^{\sum_{k=2}^{c} a-i_2\ldots-i_k} \tag{27}$$

$$= u \cdot p_1^{i_1-1}(1-p_1)^{a-i_1}(1-p_1)^{\sum_{k=2}^{c} i_k}(p_1+c-2)^{\sum_{k=2}^{c} a-i_2\ldots-i_k} \tag{28}$$

$$= u \cdot p_1^{\overbrace{i_1-1}^{\theta}} \cdot (1-p_1)^{\overbrace{a-i_1+\sum_{k=2}^{c} i_k}^{\phi}} \cdot (p_1+c-2)^{\overbrace{\sum_{k=2}^{c} a-i_2\ldots-i_k}^{\psi}}, \tag{29}$$

where $u$ is independent of $p_1$

$$u = \left( \prod_{k=1}^{c} \binom{a-i_1\ldots-i_k}{i_k} \right) (\frac{1}{c-1})^{\sum_{k=2}^{c} i_k + \sum_{k=2}^{c} a-i_2\ldots-i_k} \tag{30}$$

We observe that $\frac{f(\mathbf{i},a,\mathbf{p})}{p_1}$ demonstrates different behaviors for $a=1$ and $a>1$ and discuss the two cases separately.

(1) When $a=1$, it holds that $i_1=1$, $\theta=0$, $\phi=0$, and $\psi=0$:

$$\frac{f(\mathbf{i},a,\mathbf{p})}{p_1} = u \cdot p_1^0(1-p_1)^0(p_1+c-2)^0 = u. \tag{31}$$

In such case, $\frac{d\frac{f(\mathbf{i},a,\mathbf{p})}{p_1}}{dp_1}$ is independent with $p_1$ and remain constant when $p_1$ changes.

(2) When $a>1$, it holds that $i_1 \leq 1$, $\theta \geq 0$, $\phi \geq 0$, $\psi \geq 0$, and $\theta+\phi>0$:

$$\frac{d\frac{f(\mathbf{i},a,\mathbf{p})}{p_1}}{dp_1} = u \cdot p^{\theta-1}(1-p)^{\phi-1}(p+c-2)^{\psi-1} \cdot v, \tag{32}$$

where

$$v = \theta(1-p)(p+c-2) + \psi p(1-p) - \phi p(p+c-2) \tag{33}$$

$$= (-\theta-\phi-\psi)p^2 + (\theta+\psi+(\phi-\theta)(c-2))p + \theta(c-2). \tag{34}$$

When $0<p_1<1$ and $u>0$, $\frac{d\frac{f(\mathbf{i},a,\mathbf{p})}{p_1}}{dp_1} = 0$ if and only if $v=0$, and the corresponding value of $p_1$ is:

$$p_1^0 = \frac{-(\theta-\theta \cdot (c-2)+\phi-\psi \cdot n) - \sqrt{\Delta}}{2(-\theta-\phi-\psi)}, \tag{35}$$

$$p_1^1 = \frac{-\left(\theta - \theta \cdot (c-2) + \phi - \psi \cdot n\right) + \sqrt{\Delta}}{2\left(-\theta - \phi - \psi\right)}, \tag{36}$$

where

$$\Delta = \left(\theta - \theta \cdot (c-2) + \phi - \psi \cdot (c-2)\right)^2 - 4\left(-\theta - \phi - \psi\right)\left(\theta \cdot (c-2)\right). \tag{37}$$

It is easy to show that $\Delta > 0$, and since $(-\theta - \phi - \psi) < 0$, $v$ is concave, $v > 0$ when $p_1^1 < p < p_1^0$. Also, it is observed that when $p_1 = 0$,

$$v = \theta(c-2) \geq 0; \tag{38}$$

when $p_1 = 1$,

$$v = -\phi(c-1) < 0. \tag{39}$$

Thus it must be held that $0 < p_1^0 < 1$ and $p_1^1 \leq 0$, and for any $(\mathbf{i}, a > 1)$, there exist a $0 < p_1(\mathbf{i}, a > 1)^0 < 1$ such that when $p_1 < p_1^0(\mathbf{i}, a > 1)$, $\frac{d\frac{f(\mathbf{i}, a, \mathbf{p})}{p_1}}{dp_1} > 0$. Let

$$p_1^{\min} = \min_{\forall a \in \{2, \ldots, n'\}, \mathbf{i} \in \mathbf{I}} p_1^0(\mathbf{i}, a), \tag{40}$$

where $\mathbf{I}$ is all possible $\mathbf{i}$, then it holds that $0 < p_1^{\min} < 1$, and when $p_1 < p_1^{\min}$, $\frac{d\frac{q(a, \mathbf{p})}{p_1}}{dp_1} > 0$.

Next, we show that $\mathbb{E}[\frac{p_1'}{p_1}]$ decreases as $n'$ decreases when $p_1 < 1/c$. We first rewrite $\mathbb{E}[\frac{p_1'}{p_1}]$ as

$$\mathbb{E}[\frac{p_1'}{p_1}] = \frac{1}{p_1}\frac{1}{n'}\sum_{a=1}^{n'}\mathbb{E}[n_a]q(a, \mathbf{p}) \tag{41}$$

$$= \frac{1}{p_1}\sum_{a=1}^{n'}\binom{n-n'}{a-1}\left(\frac{1}{n'}\right)^{a-1}\left(1-\frac{1}{n'}\right)^{n-n'-(a-1)}q(a, \mathbf{p}) \tag{42}$$

First, we show that $q(a, \mathbf{p})$ is smaller for larger $a$ when $p_1 < 1/c$. The intuition is that when a new node is added to a cluster originally with $a - 1$ nodes, the new node has a higher probability of being another class, and the likelihood of class 1 becoming the majority class decreases.

Next, we show that when $n'$ increases, $\binom{n-n'}{a-1}\left(\frac{1}{n'}\right)^{a-1}\left(1-\frac{1}{n'}\right)^{n-n'-(a-1)}$ becomes more left-skewed, that it gives a smaller value for large $a$. The intuition is that, as the value of $n'$ increases, the average cluster size is expected to decrease. As a result, a large $a$ becomes farther from the average cluster size, and the probability of a cluster having exactly $a$ nodes decreases, leading to a decrease in the ratio of clusters with size $a$.

With the above observations, it can be concluded that when $p_1 < 1/c$, the value of $\mathbb{E}[\frac{p_1'}{p_1}]$ decreases as $n'$ decreases. $\qquad\square$

**Observation 4.1.** *Node Fidelity Preservation with buffer size $b$ can alleviate the declination of a minority class $k$ when $p_k$ decreases and $n'$ decreases, and prevent class $k$ from vanishing at small when $p_k$ is small.*

The mechanism of Node Fidelity Preservation is to "hold" $b$ clusters such that each cluster only has 1 node. We already show that when $a = 1$, $\frac{q(a, \mathbf{p})}{q_1}$ is independent of $q_1$ and so $\mathbb{E}[p_1'] = p_1$. By doing so, we make sure that among the $b$ nodes, class 1 does not decline as $p_1$ or $n'$ decline.

We demonstrate the effect of Node Fidelity Preservation with examples. We assign $n = 1000$, $c = 2, 3$, and $b = \lfloor n'/5 \rfloor$. we plot change of $\mathbb{E}[\frac{p_1'}{p_1}]$ and $\mathbb{E}[p_1']$ at different $n'$ and $p_1$ separately in Figure 4. We draw the trend with Node Fidelity Preservation (NFP) using dash lines and without NFP using solid lines. From the figure, we observe that without Node Fidelity Preservation being applied, the ratio $\mathbb{E}[\frac{p_1'}{p_1}]$ approaches zero when $n'$ is small, resulting in a vanishing minority class. The application of Node Fidelity Preservation prevents the ratio from approaching zero and makes sure class 1 won't disappear when $p_1$ is small.

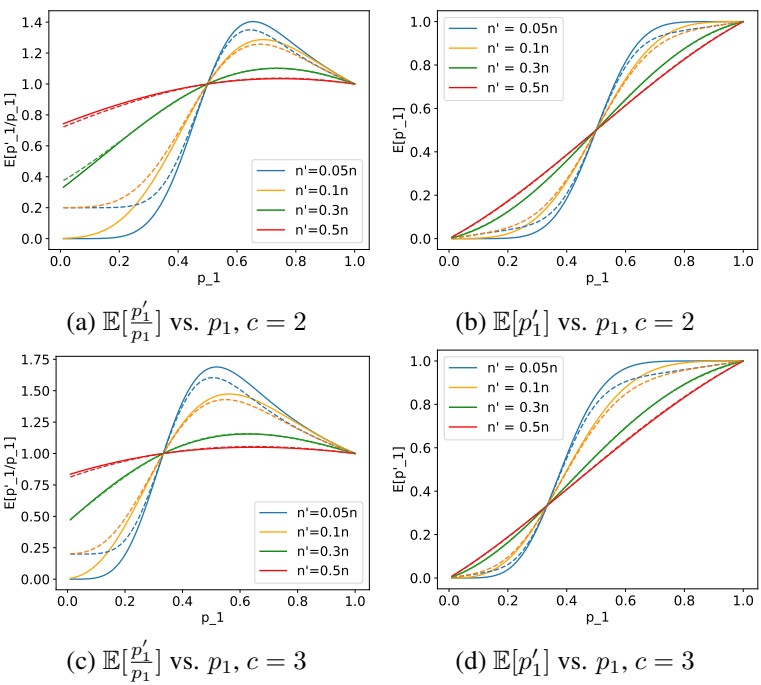

Figure 4: $\mathbb{E}[p'_1]$ and $\mathbb{E}[\frac{p'_1}{p_1}]$ against $p_1$ at different reduction rate $\gamma$ for $c = 2$ and $c = 3$. The dashed lines represent trends with Node Fidelity Preservation (NFP), and the solid lines represent trends without NFP.

### B.3    NODE REPRESENTATION PROXIMITY

Spectral-based methods aim to preserve the spectral properties of a graph. Specifically, the Laplacian matrices of the original graph and the reduced graph are compared with each other (Loukas, 2018). The combinatorial Laplacian of a graph $\mathcal{G}$, $L \in \mathbb{R}^{n \times n}$ is defined as $L = D - A$, where $A \in \mathbb{R}^{n \times n}$ is the adjacency matrix of $\mathcal{G}$, and $D \in \mathbb{R}^{n \times n}$ is its diagonal degree matrix. The combinatorial Laplacian of the reduced graph, $L' \in \mathbb{R}^{n' \times n'}$, is calculated as $L' = P^{\mp} L P^{+}$, where $P^{+}$ is the pseudo inverse of the normalized coarsening matrix $P \in \mathbb{R}^{n \times n'}$, and $P^{\mp}$ is the transposed pseudo inverse of $P$. Since $L$ and $L'$ have different sizes, they can not be directly compared. Instead, the following induced semi-norms are defined:

$$\|x\|_L = \sqrt{x^\top L x}, \ \|x'\|_{L'} = \sqrt{x'^\top L' x'}, \tag{43}$$

where $x \in \mathbb{R}^n$, and $x'$ is the projection of $x$ on $n'$-space such that $x' = Px$. The closeness between $L$ to $L'$ can be defined as how close $\|x\|_L$ is to $\|x'\|_{L'}$. $L$ and $L'$ are considered equivalent if it holds $\|x'\|_{L'} = \|x\|_L$ for any $x \in \mathbb{R}^n$. We next show that a partitioning made by merging nodes sharing the same neighborhood structure results in an equivalent Laplacian of the reduced graph as the original graph.

**Theorem B.1.** Let $i$ and $j$ be two nodes of $\mathcal{G}$, $A \in \mathbb{R}^{n \times n}$ be the adjacency matrix of $\mathcal{G}$ and $\tilde{A} = A + I$, $D$ be the diagonal matrix of $A$ and $\tilde{D} = D + I$, $P \in \mathbb{R}^{n \times n-1}$ be the normalized coarsening matrix by merging $i$ and $j$ to one node, $L$ and $L'$ be the combinatorial Laplacian of $G$ and the reduced graph. It holds that $\tilde{A}_i = \tilde{A}_j, \tilde{D}_i = \tilde{D}_j \Rightarrow \|x'\|_{L'} = \|x\|_L$ for any $x \in \mathbb{R}^n$ and $x' = Px$, where $A_i$ is the $i$-th row for a matrix A.

*Proof.* For simplicity, we assume the original graph has 3 nodes. We assume $i = 1$ and $j = 2$ are permutation-invariant. It is easy to show that the following proof can be extended to the case when the graph has more than 3 nodes. The coarsening matrix $P$ to merge the first two nodes is then:

$$P = \begin{bmatrix} \frac{1}{2} & \frac{1}{2} & 0 \\ 0 & 0 & 1 \end{bmatrix}, \tag{44}$$

then the pseudo inverse $P^+$ and the transpose pseudo inverse $P^\mp$ of $P$ is

$$P^+ = \begin{bmatrix} 1 & 0 \\ 1 & 0 \\ 0 & 0 \end{bmatrix} \qquad P^\mp = \begin{bmatrix} 1 & 1 & 0 \\ 0 & 0 & 1 \end{bmatrix}.$$

Given $L$ as the combinatorial Laplacian of the original graph, the combinatorial Laplacian $L'$ of the reduced graph is

$$L' = P^\mp L P^+, \tag{45}$$

and for a $x = [x_1, x_2, x_3]^\top$, the corresponding $x'$ is computed as:

$$x' = Px = [(x_1 + x_2)/2, x_3]^\top, \tag{46}$$

Then

$$x^\top L x = (l_{21} + l_{12})x_1 x_2 + (l_{31} + l_{13})x_1 x_3 \tag{47}$$
$$+ (l_{32} + l_{23})x_2 x_3 + l_{11}x_1^2 + l_{22}x_2^2 + l_{33}x_3^2, \tag{48}$$

$$x'^\top L' x' = \frac{1}{2}(l_{11} + l_{12} + l_{21} + l_{22})x_1 x_2 \tag{49}$$

$$+ \frac{1}{2}(l_{13} + l_{23} + l_{31} + l_{32})x_1 x_3 \tag{50}$$

$$+ \frac{1}{2}(l_{13} + l_{23} + l_{31} + l_{32})x_2 x_3 \tag{51}$$

$$+ \frac{1}{4}(l_{11} + l_{12} + l_{21} + l_{22})x_1^2 \tag{52}$$

$$+ \frac{1}{4}(l_{11} + l_{12} + l_{21} + l_{22})x_2^2 + l_{33}x_3^2. \tag{53}$$

Given $\tilde{A}_1 = \tilde{A}_2, \tilde{D}_1 = \tilde{D}_2 \Rightarrow A_1 = A_2, D_1 = D_2$ and $L = D - A$, it is straightforward to prove that $l_{1a} = l_{2a}$ and $l_{b1} = l_{b2}$ which means $l_{11} = l_{12} = l_{21} = l_{22}$ and $l_{13} = l_{23}, l_{31} = l_{32}$. Thus,

$$x^\top L x = x'^\top L' x' \tag{54}$$
$$\Rightarrow \sqrt{x^\top L x} = \sqrt{x'^\top L' x'} \tag{55}$$
$$\Rightarrow \|x'\|_{L'} = \|x\|_L. \tag{56}$$

$\square$

### B.4 PROOF OF THE SIZE OF THE REDUCED GRAPH

*Proof.* The number of nodes of the reduced graph at task $t$ is:

$$
\begin{aligned}
n &= (1 - \gamma)(...((1 - \gamma)n_1 + n_2)... + n_t) \\
&\leq (1 - \gamma)(...((1 - \gamma)n_{\text{MAX}} + n_{\text{MAX}})... + n_{\text{MAX}}) \\
&= ((1 - \gamma) + (1 - \gamma)^2 + ... + (1 - \gamma)^t)n_{\text{MAX}} \\
&\leq \frac{1 - \gamma}{\gamma}n_{\text{MAX}}
\end{aligned}
\tag{57}
$$

$\square$

### B.5 DISCUSSION ON UTILIZING SOFTLABEL AS AN ALTERNATIVE TO MAJORITY VOTING

Another potential solution to address the "vanishing class" problem caused by the majority voting is to use softlabel to represent the cluster label instead. However, such an approach may come with several drawbacks. First, using hard labels is memory-efficient, and requires only a single digit to represent a sample's label. In contrast, soft labels use a vector for storage, with a size equivalent to the model's output dimension. This distinction in memory usage is negligible for

Table 3: Statistics of datasets. "Interval" indicates the length of the time interval for each task. "# Item" indicates the total number of items/papers in each dataset.

| Dataset | Time | Interval | #Task | #Class | #Items |
|---------|------|----------|-------|--------|--------|
| Kindle | 2012-2016 | 1 year | 5 | 6 | 38,450 |
| DBLP | 1995-2014 | 2 years | 10 | 4 | 54,265 |
| ACM | 1995-2014 | 2 years | 10 | 4 | 77,130 |

models with few output classes. However, in scenarios where the model predicts among a large number of classes, the increased memory demand of soft labels becomes significant and cannot be overlooked. Secondly, although a model can learn to predict a soft label during the training phase, most applications necessitate deterministic predictions in the testing or inference phase. We are concerned that training with soft labels might lead the model towards indeterministic and ambiguous predictions, potentially undermining its practical applicability. The last concern is that when using soft labels, instead of a classification task, the model is performing a regression task. To predict single nodes with deterministic labels, it is considered a suboptimal approach to model it as a regression task due to the unaligned optimization objectives and loss functions. Also, regression is believed to be a less difficult task to learn compared to classification task as discrete optimization is generally harder than continuous optimization. Training the model with an easier task may restrict its performance during the test phase.

## C  SUPPLEMENTAL EXPERIMENT SETUPS

### C.1  DETAILS OF THE DATASETS

The Kindle dataset contains items from the Amazon Kindle store; each node representing an item is associated with a timestamp indicating its release date, and each edge indicates a "frequently co-purchased" relation between items. The DBLP and ACM datasets are citation datasets where each node represents a paper associated with its publishing date, and a node's connection to other nodes indicates a citation relation. For the Kindle dataset, we select items from six categories: *Religion & Spirituality*, *Children's eBooks*, *Health, Fitness & Dieting*, *SciFi & Fantasy*, *Business & Money*, and *Romance*. For the DBLP dataset, we select papers published in 34 venues and divide them into four classes: *Database*, *Data Mining*, *AI*, and *Computer Vision*. For the ACM dataset, we select papers published in 66 venues and divide them into four classes: *Information Systems*, *Signal Processing*, *Applied Mathematics*, and *AI*. For each of the datasets, we select nodes from a specified time period. We build a graph and make a constraint that the timestamp of the target node is not allowed to be larger than the source node, which should naturally hold for citation datasets as one can not cite a paper published in the future. Then we split each graph into subgraphs by edges based on the timestamps of the source nodes. To simulate a real-life scenario that different tasks may have different sets of class labels, at each task for the Kindle dataset, we randomly select one or two classes and mask the labels of the nodes from selected class(es) during the training and the test phase; for the DBLP and ACM datasets, we randomly select one class and mask the labels of the nodes from the selected class during the training and the test phase. The summary of each dataset is provided in Table 3.

### C.2  HYPER-PARAMETER SETTING

For each task, we randomly split all nodes into training, validation, and test sets with the ratio of $30/20/50$. For the baseline CGL models the memory strengths are searched from $\{10^i | i \in [-5...5]\}$ or $\{0.1, 0.2...0.9\}$. For baselines that utilize a memory buffer, we calibrate their memory buffer sizes to ensure that their memory usage is on a similar scale to that of **TA**ℂℚ. For **TA**ℂℚ, by default the reduction ratio is 0.5; memory buffer size for Node Fidelity Preservation is 200; node degree is used to determine a node's importance score, and Reservoir Sampling is chosen as the node sampling strategy. We chose GCN and GAT as the GNN backbones. For both of them, we set the number of layers as 2 and the hidden layer size as 48. For GAT, we set the number of heads as 8. For each dataset, we generate 10 random seeds that split the nodes into training, validation, and test sets and

select the masked class. We run all models on the same random seeds and the results are averaged over 10 runs. All experiments were conducted on a NVIDIA GeForce RTX 3090 GPU.

# D    ADDITIONAL RESULTS

## D.1    MAIN RESULTS

We present more results of the performance of **TA**ℂ𝕆 and other baselines on three datasets and three backbone GNN models, including GCN (Table 4), GAT (Table 5), and GIN (Table 6). The results show that our proposed method outperforms other continual learning approaches consistently with different GNN backbone models.

Table 4: Performance comparison of node classification in terms of F1 and BACC with GCN on three datasets (average over 10 trials). Standard deviation is denoted after $\pm$.

| Dataset | Method | F1-AP (%) | F1-AF (%) | BACC-AP (%) | BACC-AF (%) |
|---|---|---|---|---|---|
| | joint train | 87.21±0.55 | 0.45±0.25 | 80.98±0.55 | 1.17±0.82 |
| | finetune | 69.10±10.85 | 18.99±11.19 | 67.12±5.17 | 16.01±6.02 |
| Kindle | simple-reg | 68.80±10.02 | 18.21±10.49 | 66.85±4.20 | 14.79±5.09 |
| | EWC | 77.08±8.37 | 10.87±8.62 | 72.13±4.97 | 10.75±6.05 |
| | TWP | 78.90±4.71 | 8.99±4.93 | 73.16±2.94 | 9.62±3.75 |
| | OTG | 69.01±10.55 | 18.94±10.79 | 66.69±4.77 | 16.26±5.77 |
| | GEM | 76.08±6.70 | 11.01±7.27 | 71.77±2.71 | 10.13±4.14 |
| | ERGNN-rs | 77.63±3.61 | 9.64±4.19 | 70.87±3.26 | 10.83±4.23 |
| | ERGNN-rb | 75.87±6.41 | 11.46±6.98 | 71.44±2.64 | 10.98±3.64 |
| | ERGNN-mf | 77.28±5.91 | 10.15±6.31 | 72.23±3.71 | 10.27±4.74 |
| | DyGrain | 69.14±10.47 | 18.88±10.72 | 66.44±5.36 | 16.48±6.40 |
| | IncreGNN | 69.45±10.34 | 18.48±10.66 | 67.07±5.12 | 15.86±5.86 |
| | SSM | 78.99±3.13 | 8.19±3.63 | 72.95±2.60 | 8.72±2.79 |
| | SSRM | 77.37±4.06 | 9.99±4.55 | 71.46±3.46 | 10.55±4.33 |
| | **TACO** | **82.97±2.05** | **4.91±1.90** | **76.43±2.53** | **6.31±2.50** |
| | joint train | 86.33±1.38 | 0.77±0.13 | 80.55±1.45 | 1.28±0.55 |
| | finetune | 67.85±8.05 | 20.43±7.07 | 65.76±3.28 | 18.26±1.85 |
| DBLP | simple-reg | 69.70±9.16 | 18.69±8.48 | 66.99±3.94 | 17.29±2.99 |
| | EWC | 79.38±4.86 | 8.85±4.11 | 73.22±4.47 | 10.69±3.90 |
| | TWP | 80.05±3.71 | 8.23±3.28 | 73.79±3.41 | 10.44±3.53 |
| | OTG | 68.24±10.12 | 20.12±9.34 | 65.70±4.98 | 18.62±3.75 |
| | GEM | 80.04±3.24 | 7.90±2.68 | 73.01±4.58 | 10.66±4.31 |
| | ERGNN-rs | 78.02±5.79 | 10.08±5.16 | 71.70±5.11 | 12.32±4.71 |
| | ERGNN-rb | 75.16±7.24 | 12.85±6.54 | 70.26±5.16 | 13.79±4.72 |
| | ERGNN-mf | 77.42±5.25 | 10.64±4.38 | 72.26±4.21 | 11.90±3.52 |
| | DyGrain | 67.52±10.88 | 20.83±10.16 | 65.82±4.39 | 18.34±3.47 |
| | IncreGNN | 69.40±9.60 | 18.92±8.75 | 66.85±4.86 | 17.41±3.53 |
| | SSM | 82.71±1.76 | 4.20±1.26 | 76.05±2.01 | 6.21±1.68 |
| | SSRM | 77.43±5.34 | 10.66±4.47 | 71.20±4.97 | 12.76±4.27 |
| | **TACO** | **84.60±2.01** | **2.51±1.03** | **79.63±2.16** | **3.02±1.81** |
| | joint train | 75.35±1.49 | 1.87±0.60 | 66.01±1.36 | 2.35±0.99 |
| | finetune | 60.53±9.35 | 19.09±9.23 | 56.55±3.29 | 15.75±2.62 |
| ACM | simple-reg | 61.63±10.09 | 17.83±9.99 | 57.81±2.24 | 14.64±1.57 |
| | EWC | 66.48±6.43 | 12.73±6.26 | 60.70±2.00 | 11.57±1.87 |
| | TWP | 65.98±7.26 | 13.33±6.94 | 60.19±2.50 | 12.09±2.31 |
| | OTG | 61.45±9.94 | 18.33±9.86 | 57.29±2.96 | 15.07±1.90 |
| | GEM | 67.17±4.24 | 11.69±3.94 | 59.00±3.30 | 12.12±2.21 |
| | ERGNN-rs | 64.82±7.89 | 14.43±7.68 | 58.95±2.83 | 12.94±2.14 |
| | ERGNN-rb | 63.58±8.82 | 15.66±8.71 | 58.34±2.68 | 13.76±2.32 |
| | ERGNN-mf | 64.80±8.49 | 14.59±8.41 | 59.79±2.04 | 13.00±1.15 |
| | DyGrain | 61.40±9.57 | 18.47±9.50 | 57.23±3.14 | 15.39±2.08 |
| | IncreGNN | 61.32±9.70 | 18.42±9.64 | 56.97±3.23 | 15.47±2.37 |
| | SSM | 68.77±2.93 | 9.50±2.47 | 59.32±3.08 | 10.80±2.60 |
| | SSRM | 64.39±7.43 | 14.72±7.48 | 58.78±2.26 | 12.93±1.84 |
| | **TACO** | **70.96±2.68** | **8.02±2.33** | **63.98±1.67** | **7.45±2.18** |

Table 5: Performance comparison of node classification in terms of F1 and BACC with GAT on three datasets (average over 10 trials). Standard deviation is denoted after ±.

| Dataset | Method | F1-AP (%) | F1-AF (%) | BACC-AP (%) | BACC-AF (%) |
|---|---|---|---|---|---|
| | joint train | 88.54±0.70 | 0.35±0.27 | 82.71±1.02 | 0.62±0.46 |
| | finetune | 68.68±11.55 | 20.05±11.59 | 66.89±4.90 | 16.91±5.32 |
| Kindle | simple-reg | 66.20±10.89 | 19.26±11.18 | 64.61±3.98 | 15.24±4.48 |
| | EWC | 78.95±5.62 | 9.92±5.79 | 73.49±2.76 | 10.30±3.33 |
| | TWP | 78.17±6.67 | 10.85±6.71 | 73.23±3.06 | 10.78±3.58 |
| | OTG | 70.09±9.66 | 18.78±10.02 | 67.50±4.24 | 16.53±4.65 |
| | GEM | 76.46±7.14 | 11.86±7.61 | 72.52±2.56 | 10.97±3.49 |
| | ERGNN-rs | 78.64±4.36 | 9.56±4.57 | 72.19±2.95 | 10.70±3.20 |
| | ERGNN-rb | 75.60±7.14 | 12.66±7.34 | 71.86±3.16 | 11.55±3.32 |
| | ERGNN-mf | 78.14±5.22 | 10.35±5.59 | 72.94±3.05 | 10.90±3.89 |
| | DyGrain | 70.65±10.05 | 18.28±10.44 | 68.16±4.05 | 15.77±4.89 |
| | IncreGNN | 70.66±10.57 | 18.18±10.76 | 68.06±4.50 | 15.92±5.06 |
| | SSM | 81.84±2.10 | 6.58±2.59 | 74.64±3.10 | 8.81±2.62 |
| | SSRM | 78.09±4.54 | 10.20±5.15 | 71.99±2.46 | 11.17±3.39 |
| | **TACO** | **83.66±1.93** | **4.69±1.82** | **76.58±3.07** | **6.34±2.13** |
| | joint train | 83.43±1.81 | 1.08±0.31 | 76.97±1.94 | 1.79±0.64 |
| | finetune | 65.75±10.67 | 21.68±9.76 | 64.21±4.21 | 18.76±3.28 |
| DBLP | simple-reg | 68.85±9.68 | 18.49±8.58 | 66.21±4.01 | 16.81±2.75 |
| | EWC | 76.33±5.71 | 11.12±5.02 | 71.05±3.83 | 12.16±3.41 |
| | TWP | 76.64±4.47 | 10.61±3.77 | 70.95±3.22 | 12.03±2.98 |
| | OTG | 67.50±10.70 | 20.06±9.90 | 65.65±4.40 | 17.57±3.63 |
| | GEM | 73.64±6.07 | 12.76±4.77 | 67.53±4.47 | 14.42±3.24 |
| | ERGNN-rs | 75.36±5.62 | 11.87±4.62 | 70.07±3.88 | 12.95±3.44 |
| | ERGNN-rb | 71.65±7.32 | 15.20±6.34 | 67.16±3.86 | 15.37±3.07 |
| | ERGNN-mf | 74.62±6.16 | 12.55±5.29 | 68.97±4.20 | 13.90±3.61 |
| | DyGrain | 65.83±10.05 | 21.62±8.96 | 63.80±4.18 | 19.10±2.71 |
| | IncreGNN | 66.23±11.16 | 21.23±10.52 | 64.62±4.10 | 18.55±3.41 |
| | SSM | 81.47±2.48 | 4.46±1.59 | 74.62±2.41 | 6.48±1.69 |
| | SSRM | 74.31±6.00 | 12.63±4.68 | 69.15±4.18 | 13.57±3.41 |
| | **TACO** | **81.63±1.06** | **2.29±0.60** | **76.08±1.67** | **2.53±1.09** |
| | joint train | 74.89±1.53 | 1.91±0.85 | 65.81±1.40 | 2.32±1.33 |
| | finetune | 61.59±10.85 | 17.79±10.44 | 58.05±2.18 | 14.24±1.91 |
| ACM | simple-reg | 59.22±9.95 | 17.79±9.81 | 55.88±2.90 | 14.23±1.57 |
| | EWC | 66.75±7.00 | 12.20±6.93 | 61.49±1.84 | 10.44±1.75 |
| | TWP | 67.42±7.54 | 11.66±7.35 | 61.57±1.72 | 10.29±1.86 |
| | OTG | 62.24±10.88 | 16.99±11.02 | 58.55±2.37 | 13.70±1.41 |
| | GEM | 67.01±4.61 | 11.16±4.47 | 59.53±2.44 | 10.68±2.06 |
| | ERGNN-rs | 64.89±7.93 | 13.90±7.88 | 59.22±2.34 | 12.29±1.73 |
| | ERGNN-rb | 64.04±8.59 | 14.74±8.64 | 58.70±2.48 | 13.10±1.84 |
| | ERGNN-mf | 64.56±9.26 | 14.30±9.24 | 59.94±1.78 | 11.91±1.46 |
| | DyGrain | 61.66±10.58 | 17.72±10.42 | 58.15±2.49 | 14.07±1.80 |
| | IncreGNN | 62.25±10.70 | 17.22±10.60 | 58.37±2.41 | 14.01±1.55 |
| | SSM | 69.83±3.16 | 8.10±2.81 | 61.15±2.65 | 8.93±2.77 |
| | SSRM | 64.77±8.34 | 14.21±8.93 | 59.47±1.97 | 12.23±1.72 |
| | **TACO** | **70.37±2.70** | **7.64±2.43** | **62.59±1.51** | **7.52±1.94** |

Table 6: Performance comparison of node classification in terms of F1 and BACC with GIN on three datasets (average over 10 trials). Standard deviation is denoted after ±.

| Dataset | Method | F1-AP (%) | F1-AF (%) | BACC-AP (%) | BACC-AF (%) |
|---|---|---|---|---|---|
| | joint train | 84.39±0.84 | 0.55±0.34 | 77.17±0.83 | 1.45±1.07 |
| | finetune | 64.98±10.26 | 20.24±10.77 | 61.81±4.53 | 17.91±5.24 |
| | simple-reg | 65.04±10.74 | 20.04±11.13 | 62.95±4.16 | 16.62±5.14 |
| | EWC | 75.73±4.74 | 9.36±5.01 | 69.34±2.84 | 10.07±3.47 |
| | TWP | 76.14±4.35 | 9.14±4.72 | 69.12±3.06 | 10.49±3.51 |
| | OTG | 65.18±9.97 | 20.08±10.29 | 61.98±4.79 | 17.70±5.35 |
| Kindle | GEM | 72.10±6.29 | 12.40±6.80 | 66.76±2.92 | 12.05±3.67 |
| | ERGNN-rs | 73.67±3.22 | 10.83±3.84 | 66.72±3.00 | 11.84±3.40 |
| | ERGNN-rb | 70.25±7.35 | 14.39±8.01 | 65.53±3.14 | 13.70±4.38 |
| | ERGNN-mf | 72.33±6.10 | 12.47±6.64 | 67.12±3.17 | 12.28±4.27 |
| | DyGrain | 64.50±10.18 | 20.78±10.74 | 61.55±4.45 | 18.13±5.36 |
| | IncreGNN | 65.38±9.56 | 19.69±9.99 | 61.71±5.02 | 17.85±5.66 |
| | SSM | 76.47±3.37 | 8.01±3.38 | 67.78±2.84 | 10.73±3.03 |
| | SSRM | 73.75±3.25 | 10.79±3.56 | 66.55±3.25 | 12.07±3.63 |
| | **TACO** | **78.71±1.76** | **6.44±1.80** | **70.76±1.86** | **8.28±2.13** |
| | joint train | 84.42±1.47 | 1.63±0.28 | 77.69±1.07 | 2.65±0.68 |
| | finetune | 65.48±11.96 | 22.85±11.28 | 64.59±4.86 | 19.65±3.87 |
| | simple-reg | 66.84±9.64 | 21.85±8.97 | 65.04±3.77 | 19.76±2.80 |
| | EWC | 77.45±7.06 | 10.79±6.54 | 71.57±5.18 | 12.59±4.68 |
| | TWP | 77.59±4.91 | 10.64±4.44 | 71.45±4.11 | 12.75±3.81 |
| | OTG | 66.37±10.66 | 22.11±9.93 | 64.62±4.19 | 19.77±2.96 |
| DBLP | GEM | 78.71±4.45 | 9.10±3.47 | 72.24±3.86 | 11.39±3.27 |
| | ERGNN-rs | 76.63±3.94 | 11.47±3.29 | 70.45±3.51 | 13.67±3.18 |
| | ERGNN-rb | 73.23±7.29 | 14.77±6.50 | 68.79±4.52 | 15.22±3.91 |
| | ERGNN-mf | 75.96±5.74 | 12.14±4.99 | 70.74±3.55 | 13.39±3.10 |
| | DyGrain | 66.89±10.10 | 21.35±9.36 | 65.06±3.77 | 19.10±2.51 |
| | IncreGNN | 67.81±9.09 | 20.56±8.34 | 65.58±3.66 | 18.68±2.79 |
| | SSM | 80.21±7.85 | 6.74±7.29 | 73.96±5.16 | 8.24±4.99 |
| | SSRM | 76.60±4.89 | 11.55±4.30 | 70.89±4.29 | 13.30±3.90 |
| | **TACO** | **84.03±2.08** | **3.12±0.89** | **78.09±2.19** | **3.48±1.34** |
| | joint train | 71.86±1.54 | 2.99±0.74 | 62.65±1.17 | 2.98±1.60 |
| | finetune | 57.20±9.03 | 20.31±8.97 | 53.50±3.43 | 16.14±2.57 |
| | simple-reg | 57.86±9.27 | 19.52±9.02 | 53.99±3.41 | 15.95±2.21 |
| | EWC | 65.18±5.86 | 12.06±5.68 | 58.64±1.78 | 10.90±1.84 |
| | TWP | 65.45±5.56 | 11.72±5.35 | 58.76±1.78 | 10.63±1.60 |
| | OTG | 58.24±9.38 | 19.37±9.24 | 54.23±3.35 | 15.46±2.31 |
| ACM | GEM | 65.03±3.71 | 11.69±3.15 | 56.94±3.43 | 11.65±2.29 |
| | ERGNN-rs | 61.30±7.75 | 15.77±7.57 | 55.87±2.68 | 13.30±2.11 |
| | ERGNN-rb | 61.12±8.25 | 16.10±8.07 | 55.82±2.60 | 13.81±1.74 |
| | ERGNN-mf | 61.86±7.85 | 15.49±7.73 | 56.76±2.84 | 13.10±1.56 |
| | DyGrain | 58.09±9.46 | 19.43±9.27 | 54.26±3.06 | 15.36±2.45 |
| | IncreGNN | 58.21±9.17 | 19.43±9.03 | 54.10±3.15 | 15.78±2.03 |
| | SSM | 65.73±3.15 | 10.64±2.79 | 56.81±2.58 | 11.19±3.01 |
| | SSRM | 61.47±7.38 | 15.68±7.14 | 56.09±2.64 | 13.02±1.55 |
| | **TACO** | **67.19±3.12** | **9.73±2.80** | **59.06±2.40** | **9.13±2.92** |

## D.2  GRAPH COARSENING METHODS

We present the result of the performance of **TACO** with its graph coarsening module **RePro** replaced by other four widely used graph coarsening algorithms with GCN as the backbone GNN model in Table 7.

Table 7: Coarsen runtime and node classification results of **TACO** variations with different coarsening methods on three datasets with GCN (average over 10 trials). Boldface indicates the best result of each column.

| Dataset | Method | Time (s) | F1-AP (%) | F1-AF (%) | BACC-AP (%) | BACC-AF (%) |
|---|---|---|---|---|---|---|
| Kindle | Alge. JC | 8.9 | 81.09±2.15 | 6.88±2.37 | 74.48±2.70 | 8.20±2.98 |
| | Aff. GS | 65.6 | 77.42±4.44 | 10.57±4.60 | 71.43±4.02 | 11.47±4.64 |
| | Var. neigh. | 6.9 | 80.77±3.66 | 7.31±4.02 | 74.01±1.76 | 8.78±2.45 |
| | Var. edges | 10.1 | 82.29±1.95 | 5.63±2.17 | 75.13±2.67 | 7.54±2.93 |
| | FGC | 7.7 | 81.42±2.98 | 6.66±3.50 | 74.56±2.08 | 8.19±2.74 |
| | **RePro** | **2.3** | **82.97±2.05** | **4.91±1.90** | **76.43±2.53** | **6.31±2.50** |
| DBLP | Alge. JC | 70.76 | **85.33±1.19** | 2.26±0.85 | 78.32±2.31 | 4.69±2.35 |
| | Aff. GS | 237.1 | 84.34±1.70 | 3.68±1.68 | 77.15±3.31 | 6.87±3.76 |
| | Var. neigh. | 7.3 | 84.91±1.38 | 2.53±0.88 | 79.18±1.66 | 3.59±1.98 |
| | Var. edges | 28.0 | 85.13±1.86 | **2.08±0.91** | **79.92±1.61** | **2.50±0.88** |
| | FGC | 10.8 | 84.77±1.69 | 2.68±1.05 | 78.74±1.80 | 4.07±1.53 |
| | **RePro** | **1.1** | 84.60±2.01 | 2.51±1.03 | 79.63±2.16 | 3.02±1.81 |
| ACM | Alge. JC | 11.8 | 70.25±3.23 | 9.00±3.16 | 64.06±1.90 | 7.69±2.35 |
| | Aff. GS | 96.1 | 68.12±5.69 | 11.43±5.66 | 62.58±2.14 | 9.56±2.47 |
| | Var. neigh. | 10.3 | 66.83±7.53 | 12.71±7.50 | 62.25±2.15 | 10.04±2.52 |
| | Var. edges | 13.8 | **71.42±2.48** | **7.57±2.28** | **64.94±1.62** | **6.57±1.99** |
| | FGC | 7.0 | 66.97±7.44 | 12.52±7.43 | 62.70±2.01 | 9.48±2.49 |
| | **RePro** | **1.4** | 70.96±2.68 | 8.02±2.33 | 63.98±1.67 | 7.45±2.18 |

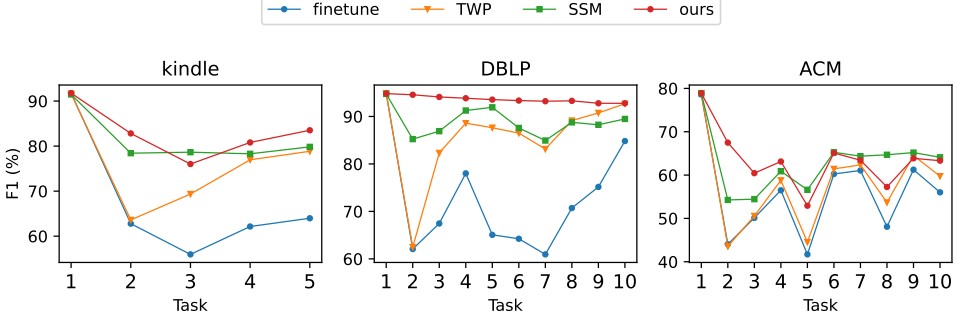

Figure 5: F1 score on the test set of the first task on Kindle, DBLP, and ACM, after training on more tasks.

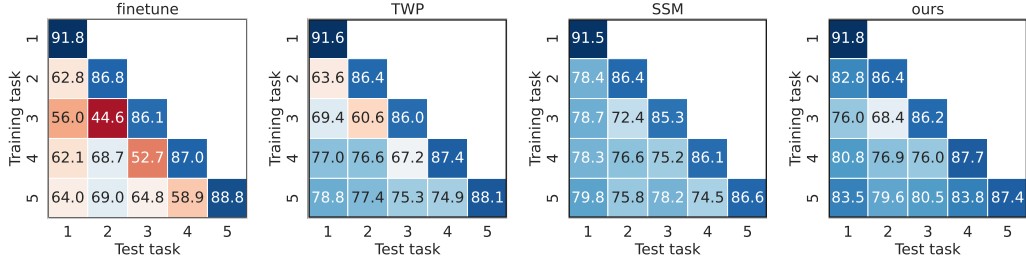

Figure 6: F1 score on the test set (x-axis) after training on each task (y-axis) on Kindle dataset.

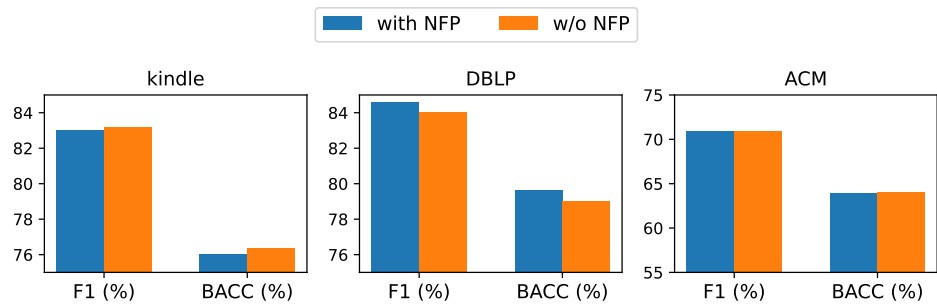

Figure 7: Average performance of the **TA**ℂ𝕆 with and without Node Fidelity Preserving (NFP).

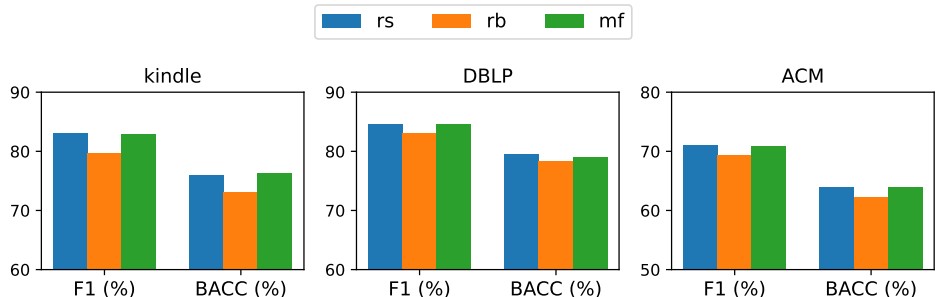

Figure 8: Average performance of the **TA**ℂ𝕆 with different node sampling strategies: Reservior-Sampling (rs), Ring Buffer (rb), and Mean Feature (mf).

## E  ADDITIONAL ABLATION STUDIES AND ANALYSIS

### E.1  PERFORMANCE AFTER TRAINING ON EACH TASK

We first investigate the performance of the model on the first task when more tasks are learned with different CGL frameworks. We report the AP-F1 of the representative baselines on the three datasets in Figure 5. It shows that with our proposed framework, the model forgets the least at the end of the last task. Also, we observe that the performance of experience replay-based methods (SSM and **TA**ℂ𝕆) are more stable through learning more tasks, but the regularization-based method, TWP, experiences more fluctuations. We deduce that this is caused by the fact that regularization-based methods can better prevent the model from drastically forgetting previous tasks even when the current task has more distinct distributions. We also observe a general pattern on the ACM dataset that the F1 score on the first task goes up after training tasks 6 and 7. Our guess is that tasks 6 and 7 share similar patterns with task 1, while the knowledge learned from task 6 and 7 is useful in predicting task 1 and thus contributes to a "positive backward transfer".

We also visualize the model's performance in terms of AP-F1 using the four approaches on all previous tasks after training on each task on Kindle dataset, as shown in Figure 6. It demonstrates that the application of different CGL methods can alleviate the catastrophic forgetting problem on the Kindle dataset as we point out in the introduction part to varying degrees.

### E.2  SHORT-TERM FORGETTING

The average forgetting (AF) measures the decline in model performance after learning all tasks, which only captures the assess the model's long-term forgetting behavior. To evaluate the short-term forgetting of the model, we introduce a new metric, termed "short-term average forgetting" (AF-st),

Table 8: The averaged short-term forgetting in terms of F1 score (%) with GCN as the backbone on three datasets (averaged over 10 trials).

| Method | Kindle | DBLP | ACM |
|---|---|---|---|
| joint train | 0.33 | 0.37 | 0.69 |
| finetune | 33.15 | 22.25 | 19.57 |
| simple-reg | 29.91 | 19.39 | 19.07 |
| EWC | 22.92 | 13.34 | 15.68 |
| TWP | 21.25 | 14.19 | 15.47 |
| OTG | 32.89 | 20.67 | 19.45 |
| GEM | 17.43 | 9.25 | 10.95 |
| ERGNN-rs | 10.42 | 8.07 | 10.23 |
| ERGNN-rb | 15.78 | 9.30 | 12.72 |
| ERGNN-mf | 13.95 | 8.72 | 13.72 |
| DyGrain | 32.76 | 20.52 | 19.67 |
| IncreGNN | 32.85 | 21.42 | 19.68 |
| SSM | 12.18 | 5.15 | 10.26 |
| SSRM | 9.68 | 8.32 | 10.51 |
| **TA**$\mathbb{CO}$ | 10.26 | 0.18 | 6.44 |

which measure the decline in model performance on the most recent task when it learns a new one:

$$\text{AF-st} = \frac{1}{T} \sum_{j=2}^{T} a_{j-1,j-1} - a_{j,j-1},$$

where $T$ is the total number of task, and $a_{i,j}$ is the prediction metric of model on test set of task $j$ after it is trained on task $i$. We report the AF-st in terms of F1 score with GCN on three datasets in Table 8.

### E.3 SHORTER TIME INTERVAL

We investigate the performance of **TA**$\mathbb{CO}$ and other baselines when each task is assigned with a shorter time interval. For the DBLP and ACM datasets, we divided them into one-year time intervals, resulting in a total of 20 tasks. We have included the AP-f1 and AF-f1 scores of all baselines utilizing GCN as their backbone in Table 9. Our findings indicate that, compared to our previous dataset splitting approach, most CGL methods exhibit a slight decline in performance, but TACO continues to outperform the other baseline models.

### E.4 EFFICIENCY ANALYSIS

We analyze the efficiency of **TA**$\mathbb{CO}$ and other experience-replay-based CGL baselines in terms of training time and memory usage. We report the averaged total training time (including the time to learn model parameters and the time to store the memory/coarsen graphs), and the averaged memory usage of the model (including the memory to store data for current task and the memory buffer to store information of previous tasks) for each task in Table 10. We find that on average, SSM uses less memory than **TA**$\mathbb{CO}$ on the Kindle dataset. However, on the DBLP and ACM datasets, SSM's memory usage is either more or similar. It's important to note that SSM maintains a sparsified graph that expands as additional tasks are introduced. As a result, SSM's memory continues to grow with the increasing number of tasks. In contrast, **TA**$\mathbb{CO}$, with its dynamic coarsening algorithm, consistently maintains a relatively stable memory regardless of the number of tasks.

### E.5 NODE FIDELITY PRESERVATION

We investigate the effectiveness of the Node Fidelity Preservation strategy by removing it from **TA**$\mathbb{CO}$ and compare the average performance of the variant of **TA**$\mathbb{CO}$ with its default version. We report the average performance of the model on the three datasets in Figure 7. We observe that on DBLP dateset, the application of Node Fidelity Preservation improves the performance, while on the other

Table 9: Node classification performance with GCN as the backbone on two datasets (averaged over 10 trials) with shorter time intervals and more tasks. Standard deviation is denoted after $\pm$.

| Method | DBLP | | ACM | |
|---|---|---|---|---|
| | F1-AP(%) | F1-AF (%) | F1-AP (%) | F1-AF (%) |
| joint train | 84.38 $\pm$1.49 | 1.60 $\pm$0.22 | 73.70 $\pm$0.71 | 3.04 $\pm$0.54 |
| finetune | 66.05 $\pm$11.30 | 22.45 $\pm$10.62 | 60.16 $\pm$8.58 | 19.56 $\pm$8.92 |
| simple-reg | 67.25 $\pm$7.80 | 21.50 $\pm$6.86 | 58.44 $\pm$7.62 | 21.10 $\pm$7.87 |
| EWC | 79.04 $\pm$6.15 | 8.84 $\pm$5.64 | 66.85 $\pm$4.66 | 11.77 $\pm$4.72 |
| TWP | 79.35 $\pm$5.83 | 8.56 $\pm$5.41 | 66.52 $\pm$4.50 | 12.15 $\pm$4.64 |
| OTG | 67.45 $\pm$8.31 | 21.03 $\pm$7.33 | 60.28 $\pm$8.18 | 19.54 $\pm$8.38 |
| GEM | 79.43 $\pm$3.66 | 8.41 $\pm$2.44 | 67.76 $\pm$3.01 | 10.58 $\pm$3.32 |
| ERGNN-rs | 75.08 $\pm$6.32 | 13.13 $\pm$5.48 | 61.43 $\pm$7.76 | 17.64 $\pm$8.16 |
| ERGNN-rb | 71.85 $\pm$7.55 | 16.46 $\pm$6.51 | 61.23 $\pm$7.67 | 18.09 $\pm$7.72 |
| ERGNN-mf | 74.24 $\pm$6.50 | 13.94 $\pm$5.37 | 63.13 $\pm$6.61 | 16.22 $\pm$6.83 |
| DyGrain | 67.96 $\pm$9.19 | 20.58 $\pm$8.35 | 61.12 $\pm$8.14 | 18.51 $\pm$8.45 |
| IncreGNN | 66.19 $\pm$7.88 | 22.34 $\pm$7.31 | 60.53 $\pm$8.42 | 19.08 $\pm$8.70 |
| SSM | 82.08 $\pm$1.99 | 4.37 $\pm$1.12 | 67.22 $\pm$1.95 | 10.63 $\pm$2.30 |
| SSRM | 75.35 $\pm$6.14 | 12.95 $\pm$5.19 | 62.11 $\pm$7.61 | 17.09 $\pm$7.91 |
| **TACO** | **83.06** $\pm$**2.25** | **4.18** $\pm$**1.69** | **68.31** $\pm$**3.21** | **10.17** $\pm$**3.63** |

Table 10: The averaged total training time (s) and the averaged memory usage (MB) for each task of experience-replay-based methods.

| Method | Kindle | | DBLP | | ACM | |
|---|---|---|---|---|---|---|
| | Time (s) | Memory (MB) | Time (s) | Memory (MB) | Time (s) | Memory (MB) |
| ERGNN-rs | 1.62 | 49.5 | 1.70 | 38.0 | 1.71 | 121.1 |
| ERGNN-rb | 0.49 | 48.5 | 0.49 | 37.5 | 0.58 | 119.4 |
| ERGNN-mf | 1.25 | 48.5 | 1.32 | 37.6 | 1.69 | 119.5 |
| DyGrain | 0.45 | 47.9 | 0.55 | 37.1 | 0.495 | 118.1 |
| IncreGNN | 0.44 | 47.9 | 0.46 | 37.1 | 0.51 | 118.1 |
| SSM | 0.89 | 53.9 | 1.32 | 48.3 | 1.35 | 144.1 |
| SSRM | 0.55 | 50.0 | 0.54 | 38.4 | 0.61 | 122.6 |
| **TACO** | 2.74 | 59.0 | 1.50 | 41.9 | 1.71 | 144.4 |

two datasets, the model performs comparably or marginally better without Node Fidelity Preservation. Note that our proposal of Node Fidelity Preservation is intended as a preventative measure, not as a means of enhancing model performance. Node Fidelity Preservation aims to prevent the situation where minority classes vanish from the reduced graph due to unbalanced class distribution. Therefore, the improvement may not be noticeable if the class distribution is relatively balanced and node degradation is insignificant. In such cases, preserving the selected nodes may prevent the graph from coarsening to the optimized structure, which could even make the performance worse

### E.6   Important node sampling strategies

We investigate how different important node sampling strategies affect the performance of the model. We report the average node classification performance of **TA**$\mathbb{CO}$ with different node sampling strategies, Reservior-Sampling, Ring Buffer, and Mean Feature on the three datasets in Figure 8. It shows that **TA**$\mathbb{CO}$ achieves better performance with Reservior-Sampling and Mean Feature. We deduce that it is caused by the fact that Ring Buffer operates on a first in, first out (FIFO) basis, that it only retains the most recent samples for each class, making it fail to preserve information from the distant past.

