# OpenReview forum: "A Topology-aware Graph Coarsening Framework for Continual Graph Learning"
_ICLR.cc/2024/Conference — Submitted to ICLR 2024_

### Official Review · Reviewer_Tw7D · 2023-10-31

**Soundness:** 3 good
**Presentation:** 3 good
**Contribution:** 3 good
**Rating:** 6
**Confidence:** 3

**Summary:**

This work investigates the problem of continual learning on time-stamped graphs. The authors propose  an edge contraction method applied to time-stamped graphs, which compresses the evolving graph to a reasonable size using node representations. At each time step, tasks are learned on the coarsened graph, and the learned representations further assist in the next time step's coarsening process. The paper also proposes Node Fidelity Preservation to retain important nodes and prevent them from being compressed. Extensive experiments have been conducted to demonstrate the effectiveness of the proposed method.

**Strengths:**

1.The motivation is clear and propsed method is quite-novel. Paper is well-written and easy to follow.
2.The experiments are thorough and comprehensive. Recently published methods have also been included in the comparisons.

**Weaknesses:**

1.In the first section, two issues related to CGL are mentioned: the problem of changing class distribution and the issue of correlation between old and new tasks. However, the subsequent chapters do not seem to clearly explain how coarsening addresses these two problems.
2.A comparison of the running time with other CGL methods is not provided.

Minors:
Final average performance is referred to as average performance.

**Questions:**

1.How does TACO address the problems raised in section 1?
2.How does TACO perform in terms of time and memory compared to previous rehearsal-based methods?

---

> ### Author Response · Authors · 2023-11-21
> **Rebuttal by Authors (part 1/1)**
>
> ## Summary:
> **A:** We appreciate your positive feedback and valuable comments. We have carefully considered each of your comments and carried additional experience in response to your suggestions. We sincerely hope our response can address your concerns.
> W1: In the first section, two issues related to CGL are mentioned: the problem of changing class distribution and the issue of correlation between old and new tasks. However, the subsequent chapters do not seem to clearly explain how coarsening addresses these two problems.
>
> ---
>
> ## W1, Q1: How TACO addresses the problem of changing class distribution and the issue of correlation between old and new tasks
> **A:**
> * **TACO addresses the catastrophic forgetting caused by the distribution shift.**  The change of class distribution is one of the reasons that caused the catastrophic forgetting: due to the distribution shift among different tasks, the model tends to forget the knowledge learned in old tasks when learning new tasks, and it would perform badly in old tasks. TACO addresses this forgetting problem by replaying experience from previous tasks in the form of reduced graphs to the model when training new tasks.
>
> * **How TACO captures the inter-correlation among tasks.** TACO captures the inter-correlation between tasks in the combination step. In the process of building subgraphs described in the problem statement, the same node may exist in both the previous timestamp and current timestamp, and they form edges with other nodes in these timestamps. Since most CGL methods treat subtasks as independent graphs, the copies of the same nodes on different graphs are unidentified, and the connections between two graphs through their overlapping nodes are lost, thus they *“fail to capture the inter-dependencies between tasks that result from the presence of overlapping nodes”.* TACO addresses this problem by aligning the overlapping nodes in old and new graphs, and successfully connects two graphs and captures the inter-correlation between tasks. We make it more clear in the methodology part of the newest version of this paper.
>
> ---
>
> ## W2, Q2: Running time and memory analysis
> **A:** Per your recommendation, we have conducted an efficiency analysis of TACO and other rehearsal-based CGL baselines in terms of training time and memory usage. We report the averaged total training time (including the time to learn model parameters and the time to store the memory/coarsen graphs), and the averaged memory usage of the model (including the memory to store data for the current task and the memory buffer to store information from previous tasks) for each task in Table 10.
>
> We include the discussion in Appendix E.4 in the newest version of this paper.

---

> > ### Comment · Reviewer_Tw7D · 2023-11-21
> > **Thanks for the authors' reply and i could like to retain my score.**
> >
> > Thanks for the authors' reply. My concerns have been addressed and also incorporated the response into the manuscripts. I retain my score.

---

### Official Review · Reviewer_A9ak · 2023-10-31

**Soundness:** 3 good
**Presentation:** 3 good
**Contribution:** 3 good
**Rating:** 8
**Confidence:** 3

**Summary:**

The paper presents a framework called TACO (Topology-Aware Coarsening) for continual graph learning (CGL). The primary focus is on addressing the challenges of catastrophic forgetting and inefficiency in graph neural networks (GNNs) when dealing with streaming graph data. TACO aims to preserve topological information from previous tasks in a reduced graph, which is then used for training on new tasks. The paper also introduces a graph reduction algorithm called RePro, designed to efficiently reduce the size of the graph while preserving its topological properties. The authors validate their approach through experiments on real-world datasets using different GNN backbones.

**Strengths:**

- **Significance:** The paper addresses a significant gap in the literature by focusing on continual learning in the context of graph data, which is less explored compared to Euclidean data like images and text. The proposed framework achieves SOTA results on three different online datasets. The author also included detailed ablation studies on the proposed fidelity preserving and important node sampling.
- **Originality**: Using a coarse-grained representation as a buffer for continual learning is very interesting and novel.
- **Clarity**: The method explanation is very clear, and there is a nice visual representation and pseudocode to illustrate the method.

**Weaknesses:**

- Additional experiments comparing the computational efficiency (training time and inference time) of TACO with other methods could strengthen the paper.
- The paper primarily focuses on academic datasets for validation. It would be beneficial to see how TACO performs in more practical applications such as recommendation systems.
- I am a little worried that the hyperparameters for baseline methods are not well-tuned. For example, the optimal learning rate and batch size could be different for different baselines.

**Questions:**

- The author argues that using majority voting to assign a class label to a super node would result in the gradual loss of minority classes. However, in my opinion, a simple solution to this issue is to treat the super node label as a soft label, allowing for the distribution of labels within the super node. Would this approach still have the same issue in terms of the representation power? I hope the author could clarify a bit on this point.
- Will the proposed method be affected by the time interval of the dataset?
- The author utilized average forgetting as the metric, which is defined as the maximum forgetting observed thus far. It would be beneficial if the author could also compare long-term forgetting and short-term forgetting with the baseline. This comparison is relevant because various continual learning practices may have distinct requirements.

**Minor comments**:

- In Appendix A, "GraphCoarseninAlgorithm" should be "GraphCoarseningAlgorithm".

---

> ### Author Response · Authors · 2023-11-21
> **Rebuttal by Authors (part 1/2)**
>
> ## Summary:
> We are grateful for your positive feedback and the valuable insights you provided regarding our paper. We have given careful consideration to each of your comments and have carried out additional experiments to address your questions. We sincerely hope that our responses have effectively addressed any concerns you may have had.
>
>  ---
>
> ## W1: Additional computational efficiency analysis
> **A:** Per your recommendation, we have conducted an efficiency analysis of TACO and other rehearsal-based CGL baselines in terms of training time and memory usage. We report the averaged total training time (including the time to learn model parameters and the time to store the memory/coarsen graphs), and the averaged memory usage of the model (including the memory to store data for the current task and the memory buffer to store information from previous tasks) for each task in Table 10.
>
> It is worth noting that since all CGL methods are backboned with the same GNN models, the differences between these methods primarily affect the training process of the GNNs and do not impact the inference phase. Consequently, we anticipate that all CGL methods will have similar inference times.
>
> We include the discussion in Appendix E.4 in the newest version of this paper.
>
> ---
>
> ## W2: More applications of TACO
> **A:** We conducted an evaluation of TACO on citation datasets as well as an Amazon Kindle co-purchasing dataset. At present, TACO primarily emphasizes the node classification task. We concur with the reviewer's suggestion that broadening the applicability of TACO to encompass a wider array of tasks, such as recommendation systems and link prediction, would be advantageous. We intend to address this aspect in our future work and will include a discussion of these extensions in our forthcoming research.
>
>  ---
>
> ## W3: Hyper-parameter tuning of baselines
> **A:** We understand the reviewer’s concern about the hyperparameters tuning of the baselines. The GNN model hyperparameters for the backbone of CGL baselines are the same. When training a GNN, at each batch we update the model on the whole graph, so the batch size is not a tunable parameter here. For other hyperparameters of the CGL models, we took the reported value from their original papers as reference, and carefully searched for an optimized value from a wide range of values. For example, the memory strength is an important hyperparameter that decides the learning rates of the CGL baseline and controls the ratios between learning new tasks and consolidating old tasks. We notice its reported default value can have different scales for different CGL methods. Thus, we searched their values in  {10^i|i ∈ [−5...5]} and {0.1, 0.2...0.9} and selected optimal value on the validation sets.
>
>  ---
>
> ## Q1: Using soft label an alternative of majority voting
> **A:** I agree with you that assigning soft labels would be a simple solution to mitigate the vanishing minority class problem. However, we also have concerns that such an approach may come with several drawbacks. First, using hard labels is memory-efficient, and requires only a single digit to represent a sample's label. In contrast, soft labels use a vector for storage, with a size equivalent to the model's output dimension. This distinction in memory usage is negligible for models with few output classes. However, in scenarios where the model predicts among a large number of classes, the increased memory demand of soft labels becomes significant and cannot be overlooked.
>
> Secondly, although a model can learn to predict a soft label during the training phase, most applications necessitate deterministic predictions in the testing or inference phase. We are concerned that training with soft labels might lead the model towards indeterministic and ambiguous predictions, potentially undermining its practical applicability.
>
> The last concern is that when using soft labels, instead of a classification task, the model is performing a regression task. To predict single nodes with deterministic labels, it is considered a suboptimal approach to model it as a regression task due to the unaligned optimization objectives and loss functions. Also, regression is believed to be a less difficult task to learn compared to classification task as discrete optimization is generally harder than continuous optimization. Training the model with an easier task may restrict its performance during the test phase.
>
> With the above concerns, we still believe majority voting is a reliable and convenient method to obtain the labels of super-nodes. Also, majority voting has been used in scalable graph learning (Huang et al., 2021). Nevertheless, we appreciate your thoughtful consideration and insightful view of this problem. We will add this discussion to the newest version of our paper and conduct empirical studies for this approach in the future.

---

> ### Author Response · Authors · 2023-11-21
> **Rebuttal by Authors (part 2/2)**
>
> ## Q2: Will the proposed method be affected by the time interval of the dataset?
>
> **A:** We appreciate your bringing up this significant issue for study. In response to your query, we have conducted an investigation into the performance of TACO and other baseline models when tasks are allocated shorter time intervals. For the DBLP and ACM datasets, we divided them into one-year time intervals, resulting in a total of 20 tasks. We have included the AP-f1 and AF-f1 scores of all baselines utilizing GCN as their backbone in Table 9. Our findings indicate that, compared to our previous dataset splitting approach, most CGL methods exhibit a slight decline in performance. but TACO continues to outperform the other baseline models.
>
> Regarding the Amazon-Kindle dataset, as we have already divided the tasks based on the finest available granularity, we are unable to further reduce the time interval for each task.
>
> We include the discussion in Appendix E.3 of the latest version of our paper.
>
> ---
>
> ## Q3: Short-term forgetting
>
> **A:**  We concur with your observation that the forgetting matrices we have previously applied primarily assess the model's long-term forgetting behavior. In response to your suggestion, we have introduced a new metric, termed "short-term average forgetting" (AF-st), which measure the decline in model performance on the most recent task when it learns a new one. AF-st is formally defined as follows:
>
> \begin{aligned}
>  \text{AF-st} = \frac{1}{T} \sum_{j=2}^T{{a_{j-1,j-1}-a_{j,j-1}}}.
> \end{aligned}
>
> Here, $T$ is the total number of tasks, and $a_{i,j}$ is the prediction metric of the model on the test set of task $j$ after training on task $i$. We report the AF-st in terms of F1 score using GCN on three datasets in Table 8. We include the discussion in Appendix E.2 of the latest version of our paper.
>
> Additionally, in Appendix E.1, we have provided visualizations of the performance of TACO and three representative baselines on each test set after training on each task within the Amazon-Kindle dataset. These visualizations also serve to illustrate the short-term forgetting patterns exhibited by each model to our readers.
>
>
> ---
>
> ## Minor issues:
> **A:** We appreciate you pointing out the spelling mistake. We have now corrected this typo and conducted a thorough proofreading of the paper again.

---

> ### Author Response · Authors · 2023-11-22
>
> Dear Reviewer A9ak
>
> Thank you once more for your valuable feedback. If you have any additional questions or concerns, please feel free to share them with us before the end of the author discussion period. We are eager to address any further queries during our discussion.
>
> Thank you!

---

### Official Review · Reviewer_sD2m · 2023-11-01

**Soundness:** 3 good
**Presentation:** 2 fair
**Contribution:** 2 fair
**Rating:** 5
**Confidence:** 4

**Summary:**

This paper introduce a graph-coarsening based method for continual graph learning. To avoid the forgetting problem, the proposed method would reduce the learnt graph part by graph coarsening and connect it to the new task graphs, so that the learning on new tasks would also take old information into consideration. The main part of the paper is the coarsening algorithm, while the other part largely follow existing works.

**Strengths:**

1. continual graph learning is a more practical than static graph learning setting, and should be paied more attention.
2. The proposed method outperforms the baselines.

**Weaknesses:**

1. The literature review is very incomplete.
2. Due to point 1, it is hard to position the paper against the literature and find out what is the contribution.
3. The dataset splitting is problematic.

**Questions:**

1. It is problematic to claim that the existing work focus on task-incremental-learning and sequential tasks are independent graphs. First, many benchmark works including [1,2] study the class-incremental-learning. Second, not all continual graph learning works study the independent graphs. For example, [3] does not split the growing graphs into independent graphs, [4] may also be a related work. Similar works are abundant and the authors are enouraged to do a thorought literature review.

2. The datasets are split according to time, then how do the classes increment? Does each time stamp necessarily contain new classes?

3. According to Table 3 in the appendix, the number of classes are very small. For DBLP and ACM, 4 classes are used to construct 10 tasks. Then many tasks are actually containing same classes, and it is not a class-incremental situation with large distribution gap.


4. Is the proposed method related to graph pooling methods? Graph pooling methods seem to be able to maintain a low computation burden while considering the node features at the same time.

[1] Carta, Antonio, et al. "Catastrophic forgetting in deep graph networks: an introductory benchmark for graph classification." arXiv preprint arXiv:2103.11750 (2021).

[2] Zhang, Xikun, Dongjin Song, and Dacheng Tao. "Cglb: Benchmark tasks for continual graph learning." Advances in Neural Information Processing Systems 35 (2022): 13006-13021.

[3] Feng, Yutong, Jianwen Jiang, and Yue Gao. "Incremental Learning on Growing Graphs." (2020).

[4] Das, Bishwadeep, and Elvin Isufi. "Graph filtering over expanding graphs." 2022 IEEE Data Science and Learning Workshop (DSLW). IEEE, 2022.

---

> ### Author Response · Authors · 2023-11-16
> **Rebuttal by Authors (part 1/2)**
>
> ## Summary:
>
> We appreciate your insightful suggestions. We hope our answers can address your concerns regarding the thoroughness of our literature review, the implications for the perceived contribution of our work, and the validity of our class-incremental learning setting.
>
> ---
> ## W1, W2, Q1: completeness of the literature review, and the contribution of this work
>
> **A:**
> * **Clarification of our claim.** First, we would like to clarify that we are targeting at both class-incremental and an inductive setting, which is challenging. We found most existing works are either not class-incremental or not inductive [1] [2]. We acknowledge that our original expression could be misleading, and we will clarify it in the newest version.
>
> * **Discussion on related work [1][2].** We value your contribution in providing [1][2] as benchmarks, and we intend to incorporate them into the discussion in the latest version of this paper. However, it's worth noting that in [1][2], the tasks are split based on classes rather than time steps, making them substantially different than our goal and less suitable for integration into our experiments.
>
> * **Discussion on related work [3].** We agree that [3] is a related work and it does not split the growing graphs into independent graphs. However, as we mentioned in the related work, [3] studies the Dynamic Graph Learning that *“focuses on learning up-to-date representations on dynamic graphs instead of tackling the forgetting problem. They assume the model has access to previous information. The challenges explored in the DGL fundamentally differ from those addressed in our paper.”*
>
> * **Discussion on related work [4].**  It seems that the goal of [4] is to denoise an expanding graph. We kindly ask the reviewer to elucidate the connection or similarity between [4] and our work, and we are willing to add [4] to the discussion of our paper.
>
> * **Contribution and novelty of our work.** We would like to explicitly clarify that the core contribution of our work lies in the development of a dynamic and efficient graph coarsening framework. The framework is designed to adeptly address the challenges of preserving graph topology and inter-task correlation within an experience-replay-based Continual Graph Learning (CGL) setting. **We assert that our contributions remain unchallenged by any existing literature we have encountered, and the works mentioned by the reviewers.** Also, our approach has been recognized for its novelty by reviewers 9rSM, A9ak, and Tw7D.
>
> * **Completeness of literature review.** In preparing our work, we conducted extensive literature reviews of existing state-of-the-art CGL methods, particularly those centered on experience replay and targeted the challenges we address. However, we are open to and welcome any additional literature the reviewer might suggest that addresses the same challenges or employs similar ideas.
>
> ---
> ## W3, Q2, Q3: problems of dataset splitting and class-incremental setting
> **A:**
> * **Traditional class-incremental setting.** We agree with you that our setup is different from the traditional class-incremental setting.  As we mentioned, we observe the most existing work split tasks by different classes, thus each task would have a distinct set of classes, and a new task must contain new classes.
> * **Our class-incremental setting.** However, in our split-by-time setup, it is very likely that the tasks have overlapping classes among each other. As we demonstrate in the introduction part, the class distribution across time shifts. Moreover, “in real-life situations, tasks may have different class sets (e.g. new fields of study emerge and old fields of study become obsolete)”. Thus, as mentioned in the paper, we simulate a real-life scenario that different tasks may have different sets of class labels; at each task, we randomly select one or two classes and mask the labels of the nodes from selected class(s) during the training and the testing phase. Thus, with our setup, a new task does not necessarily contain new classes that have never appeared before; instead, previously encountered classes may reappear in new tasks, and same classes can be shared across tasks.
>
> * **Generalized Class-Incremental setting.** To this end, we clarify that our experimental setup is more appropriately characterized as a Generalized Class-Incremental setting (Mi et al., 2020). In this setting, tasks do not necessarily have entirely distinct class sets. Instead, same classes can be shared across tasks, and previously encountered classes may reappear in new tasks. This setting precisely matches the class-incremental setting we use. We will clarify this in the newest version of this paper.

---

> > ### Author Response · Authors · 2023-11-16
> > **Rebuttal by Authors (part 2/2)**
> >
> > ## Q4: Connection between our work and graph pooling methods
> >
> > **A:** According to Pang et al. (2021), Liu et al. (2023) and Lee et al. (2019), the objective of graph pooling is to obtain a graph-level representation for the whole graph, and graph pooling methods can be grouped into the following three categories: topology based, global, and hierarchical pooling. Graph coarsening algorithms are often used in topology-based graph pooling. Thus, graph coarsening can be one of the techniques used in graph pooling. In this paper, our goal is to obtain a reduced version of graph with preserved graph properties, which is substantially different from graph pooling. Nevertheless, we appreciate the reviewer's insight highlighting the connection between graph pooling and the graph coarsening methods employed in our research.  We will include graph pooling in the discussion of graph coarsening in the newest version of this paper.

---

> ### Author Response · Authors · 2023-11-22
>
> Dear Reviewer sD2m
>
> We would like to thank you for your time and effort in reviewing our work. Your feedback on our responses has been very helpful in improving our paper. We hope we've managed to address the concerns you raised. If there's anything else you would like to discuss, please feel free to let us know before the author discussion period closes. We're here to respond to any further questions during the discussion.
>
> Thank you!

---

> ### Comment · Reviewer_sD2m · 2023-11-23
> **response to the rebuttal**
>
> Thanks for the detailed responsed provided by the authors.
>
> I appreciate that the authors have made revision to improve their paper. After reading the rebuttal, my main concern is on the dataset splitting and task construction.
>
> The authors claim that the continual learning scenario is the real world scenario would be that the classes in different tasks have overlap, which is true. However, constructing the tasks in this way has certain problems.
>
> When splitting the datasets into tasks, the classes are randomly chosen. However, different combinations of the classes and different orders may causes significant difference in the task difficulty. For example, when the class overlap between tasks is large, then continually learning on these tasks may not cause significant forgetting issue, and it would be hard to check whether the applied method can indeed resolve the forgetting issue.
>
> In traditional splitting, in which the tasks are constructed with different classes, it not only better ensure that different tasks follow different distribution, but also make it easier for different papers to follow a consistent setting (or different papers may have different randomly constructed tasks, with different learning difficulty).
>
> The other concerns are mostly resolved, and I could increase my rating to 5. Since the experimental setting is an important issue, I'm afraid that I would not increase the rating to a positive level.

---

### Official Review · Reviewer_9rSM · 2023-11-05

**Soundness:** 3 good
**Presentation:** 3 good
**Contribution:** 2 fair
**Rating:** 3
**Confidence:** 5

**Summary:**

TACO tackles graph continual learning (node classification) under streaming (time-stamped) graph settings. As a rehearsal-based method, it preserves graph topology information and captures the correlations between two tasks. To preserve graph topology information, a graph coarsening algorithm based on Node Representation Proximity (RePro) serves as the main component in the framework. RePro leverages node embedding from the first GNN layers and calculates the cosine similarity for connected nodes and merges nodes with high similarity, which satisfy the requirements of feature similarity, neighbor similarity and geometry closeness.  It also preserves representative nodes in the buffer for keeping important path information. Using GCN as backbones, it demonstrates the effectiveness by showing better experimental results using CL methods such as regularized-based methods (EWC,TWP…) and rehearsal-based methods(ER variants) on three datasets. It also provides experimental comparisons between different graph algorithms.

**Strengths:**

* Combining graph coarsening into task-incremental online graph continual learning is novel.

* Have very detailed algorithms and theoretical analysis.

* The experiments are comprehensive and the ablation study is provided.

**Weaknesses:**

* The introduction section contains inaccurate information that it is insufficient to categorize existing common CGL methods into only two categories: regularization-based and rehearsal-based in the Introduction section. Parametric isolation-based methods should also be mentioned.(Similarly issues in the related work)  Moreover, putting a Kindle e-book co-purchasing network showcase in the Intro sections is unnecessary (it’s just an example class incremental setting) and has no logistic connection for existing problems in CGL.

* The problem setting is unclear. One natural way for class-incremental setting is the graph expanding setting(new class and new nodes come in), causing distribution shifting. However, in this problem setting, we only observe subgraphs for each task. Does it mean pre-existing nodes can disappear? This setting seems unrealistic.

* The explanation of merging supernodes is not clear. How exactly do two nodes merge?

* This method also stores representative nodes but the strategies it uses: Reservoir Sampling: randomly sampling; ring buffer: FIFO manner and MoF. The claim of representative nodes could not hold. Which one used in your experiment is not stated.

* Compared with Var.neigh/edges, experiment gains for RePro are marginal

**Questions:**

GCN only works for graphs with fixed numbers across tasks. Choosing this model as a backbone assumes we already know how many nodes are in all tasks. This is not realistic in a continual learning setting.

---

> ### Author Response · Authors · 2023-11-16
> **Rebuttal by Authors (part 1/2)**
>
> ## Summary
>
> We value your insightful suggestions, and we appreciate your endorsement of the novelty, evaluation comprehensiveness, and technical soundness of our paper. We have carefully considered each of your comments. We sincerely hope our answers address your concerns.
>
> ---
> ## W1-1: Common CGL methods
>
> **A:** We agree with you that Parametric isolation-based (same as expansion-based) methods are also one of the approaches in Continual Learning literature, as we mentioned in the Related work: *“Regularization, Expansion, and Rehearsal are common approaches to overcome the catastrophic forgetting problem (Tang & Matteson, 2020) in continual learning”.* We observed most expansion-based methods (Rusu et al., 2016; Yoon et al., 2017; Jerfel et al., 2018; Lee et al., 2020) are not tailored to graph-structured learning.  When we did literature review for this paper, we found very limited expansion-based methods in CGL literature, and we made the observation that *“most existing CGL methods adapt regularization and rehearsal methods on graphs”.* As suggested by the reviewer, we found a recently published paper (Zhang et al., 2023) that proposed a parametric isolation approach for CGL. We change the statement and add this paper to the discussion in the newest version of our paper.
>
> ---
> ## W1-2 and W2: Problem setting and the necessity of Kindle showcase
>
> **A:**
> * **Incremental learning setting.** Most CGL papers split data into different tasks by their labels (which can be done in either a class-incremental manner, or a task-incremental manner, with their differences explained in the introduction part). In such cases, it’s more obvious that each task would have completely different class distributions, and the models tend to forget the previously learned knowledge drastically. However, we argue such a setting is not very realistic. As we mentioned in the introduction part, *“Real-world graphs often form in an evolving manner, where nodes and edges are associated with a time stamp indicating their appearing time, and graphs keep expanding with new nodes and edges. For instance, in a citation network, each node representing a paper cites (forms an edge with) other papers when it is published. Each year more papers are published, and the citation graph also grows rapidly. **In such cases, it is necessary to train a model incrementally and dynamically because saving or retraining the model on the full graph can be prohibitively expensive in space and time.”*** Thus, the pre-existing nodes will not necessarily “disappear”, but the model would have no access to them due to the cost in storing their information.
>
>  * **Necessity of using Kindle showcase in the introduction.**  We use the showcase to explain the motivation of this paper. Since we split the full graph into different tasks by time, instead of class labels, it became less obvious that nodes from different tasks may have different distributions. Thus, we believe it is necessary to demonstrate to readers that such distribution shift may happen across time, and catastrophic forgetting on old tasks exists when model learn new tasks.
>
> ---
> ## W3: The process of merging  supernodes
> **A:** The process of merging nodes into super-nodes is the same process as partitioning the nodes in a graph into clusters. When two nodes are merged, they are assigned to the same cluster (or the super-node). As it is defined in Problem Statement- Graph Coarsening, a matrix P is used to represent the node partitioning assignment of a graph. The adjacency matrix of the reduced graph, the features, and the labels of the super-nodes are decided by equation (3). We make it clearer in the methodology part to eliminate the confusion in the newest version.
>
> ---
> ## W4: Node sampling strategies
> **A:** The motivation of the node fidelity preservation is to overcome the “vanishing minority class” problem by making sure certain nodes won’t be merged to a cluster. We apply different strategies that may select nodes with different characteristics and are representative in different manners.  Reservoir Sampling randomly selects nodes from all nodes across all time, and it tends to preserve the class distribution. Ring buffer represents the most recent samples from each class. MoF selects the nodes that are closer to the average feature of each class. All three are commonly used sampling strategies and are implemented following Chaudhry et al. (2019).
>
> In appendix c.2, we mentioned that *“Reservoir Sampling is chosen as the node sampling strategy”*. Also, we perform ablation studies on the effect of different sampling strategies in E.6.

---

> > ### Author Response · Authors · 2023-11-16
> > **Rebuttal by Authors (part 2/2)**
> >
> > ## W5: Compared with Var.neigh/edges, experiment gains for RePro are marginal
> >
> >
> > **A:** We agree with you that the experiment gains of RePro are marginal compared with Var.neigh/edges in terms of AP-F1. Nonetheless, our principal aim with RePro is to offer a time-efficient solution relative to current state-of-the-art spectral-based graph coarsening methods, including Var.neigh/edges.  Our primary objective is to compete with the baselines graph coarsening method in terms of running time instead of prediction performance. As we mentioned in the methodology part: *“However, estimating spectral similarity between two nodes is typically time-consuming, even with approximation algorithms, making it less scalable for our applications where graphs are dynamically expanded and coarsened. Thus, we aim to develop a more time-efficient algorithm (RePro) that considers the aforementioned similarity measures.”* As it demonstrates in Table 2, we observe that *“RePro achieves better or comparable prediction performance compared with the best baseline while being considerably more efficient in computing time compared to all other models.”*
> >
> >  ---
> > ## Q1: GCN only works for graphs with fixed numbers across tasks.
> >
> > **A:** In addressing your concern, we would like to clarify a potential misunderstanding regarding the functionality of Graph Convolutional Networks (GCN) in relation to graph size.  With the common implementation of most GNNs (including GCN, GAT, GIN, etc.), the number of model parameters of the model is independent of the number of nodes and graph structures in a node classification task. Thus, the same GNN can run on different graphs with unknown and potentially different graph sizes and structures.

---

> ### Author Response · Authors · 2023-11-22
>
> Dear Reviewer 9rSM
>
> Thank you once again for your thorough reviews and insightful feedback. We appreciate the time and effort you've dedicated to reviewing our paper. Your comments have been helpful in enhancing the paper's quality, and we've tried our best to address your concerns in the paper. If there are any further concerns or questions, please feel free to let us know before the author discussion period ends. We will be happy to answer them during the discussion.
>
> Thank you!

---

### Meta-Review · Area_Chair_hpCb · 2023-12-07

**Metareview:**

This paper tackles the problem of catastrophic forgeting in continual graph learning. The solution relies on a graph coarsening technique that is aimed to be efficient and preserve graph topology.

The reviewers found the idea of using graph coarsening for continual graph learning to be interesting and novel. Generally the paper is well structured and there is ueful theoretical analysis as part of the paper. To further shed light into the method, the authors have also provided an ablation study.

On the other hand, there have been concerns around the insufficiency of the literature review and need for experimental comparisons with more baselines. These two issues make it difficult to understand the actual merit of the method and place it in context of the rest of the literature.

Further, one reviewer has concerns about the splitting of the dataset which I think have not been addressed satisfactorily: the splitting setting used is rather unusual which either hinders the generality of the method (if it only works in that setting) or it means that more experiments are needed to compare with papers where different settings are used.

I was also wondering about the results presented in Table 10 post rebuttal, which show unconvincing results for efficiency. This is curious since the authors claim that time-efficiency is a key goal of their approach.

As a less important point, there have been several other clarification questions by the reviewers which have been largely addressed in the rebuttal, but mainly through additions to the Appendix. I find that the updated version of the paper is still not self-contained enough.

**Justification For Why Not Higher Score:**

The paper is interesting but various issues remain around convincingness of the merit of the method compared to others in the literature, especially with regard to the splitting setting used in the continual learning. Further, even with the post-rebuttal clarifications the new revision is not self-contained enough and needs further work.

**Justification For Why Not Lower Score:**

N/A

---

### Decision · Program_Chairs · 2024-01-16

Reject